# OBSERVATIONAL OVERFITTING IN REINFORCEMENT LEARNING

**Xingyou Song**[∗]**, Yiding Jiang**[†]**, Stephen Tu, Behnam Neyshabur**
Google
{xingyousong,ydjiang,stephentu,neyshabur}@google.com

**Yilun Du**[∗]
MIT
yilundu@mit.edu

## ABSTRACT

A major component of overfitting in model-free reinforcement learning (RL) involves the case where the agent may mistakenly correlate reward with certain spurious features from the observations generated by the Markov Decision Process (MDP). We provide a general framework for analyzing this scenario, which we use to design multiple synthetic benchmarks from only modifying the observation space of an MDP. When an agent overfits to different observation spaces even if the underlying MDP dynamics is fixed, we term this *observational overfitting*. Our experiments expose intriguing properties especially with regards to *implicit regularization*, and also corroborate results from previous works in RL generalization and supervised learning (SL).

## 1 INTRODUCTION

Generalization for RL has recently grown to be an important topic for agents to perform well in unseen environments. Complication arises when the dynamics of the environments entangle with the observation, which is often a high-dimensional projection of the true latent state. One particular framework, which we denote by *zero-shot supervised framework* (Zhang et al., 2018a;c; Nichol et al., 2018; Justesen et al., 2018) and is used to study RL generalization, is to treat it analogous to a classical supervised learning (SL) problem – i.e. assume there exists a distribution of MDP's, train jointly on a finite "training set" sampled from this distribution, and check expected performance on the entire distribution, with the fixed trained policy. In this framework, there is a spectrum of analysis, ranging from almost purely theoretical analysis (Wang et al., 2019; Asadi et al., 2018) to full empirical results on diverse environments (Zhang et al., 2018c; Packer et al., 2018).

However, there is a lack of analysis in the middle of this spectrum. On the theoretical side, previous work do not provide analysis for the case when the underlying MDP is relatively complex and requires the policy to be a non-linear function approximator such as a neural network. On the empirical side, there is no common ground between recently proposed empirical benchmarks. This is partially caused by multiple confounding factors for RL generalization that can be hard to identify and separate. For instance, an agent can overfit to the MDP dynamics of the training set, such as for control in Mujoco (Pinto et al., 2017; Rajeswaran et al., 2017). In other cases, an RNN-based policy can overfit to maze-like tasks in exploration (Zhang et al., 2018c), or even exploit determinism and avoid using observations (Bellemare et al., 2012; Machado et al., 2018). Furthermore, various hyperparameters such as the batch-size in SGD (Smith et al., 2018), choice of optimizer (Kingma & Ba, 2014), discount factor $\gamma$ (Jiang et al., 2015) and regularizations such as entropy (Ahmed et al., 2018) and weight norms (Cobbe et al., 2018) can also affect generalization.

---

[∗]Work partially performed as an OpenAI Fellow.
[†]Work performed during the Google AI Residency Program. http://g.co/airesidency

Due to these confounding factors, it can be unclear what parts of the MDP or policy are actually contributing to overfitting or generalization in a principled manner, especially in empirical studies with newly proposed benchmarks. In order to isolate these factors, we study one broad factor affecting generalization that is most correlated with themes in SL, specifically *observational overfitting*, where an agent overfits due to properties of the observation which are irrelevant to the latent dynamics of the MDP family. To study this factor, we fix a single underlying MDP's dynamics and generate a distribution of MDP's by only modifying the observational outputs.

Our contributions in this paper are the following:

1. We discuss realistic instances where observational overfitting may occur and its difference from other confounding factors, and design a parametric theoretical framework to induce observational overfitting that can be applied to *any* underlying MDP.
2. We study observational overfitting with linear quadratic regulators (LQR) in a synthetic environment and neural networks such as multi-layer perceptrons (MLPs) and convolutions in classic Gym environments. A primary novel result we demonstrate for all cases is that *implicit regularization* occurs in this setting in RL. We further test the implicit regularization hypothesis on the benchmark CoinRun from using MLPs, even when the underlying MDP dynamics are changing per level.
3. In the Appendix, we expand upon previous experiments by including full training curve and hyperparamters. We also provide an extensive analysis of the convex one-step LQR case under the observational overfitting regime, showing that under Gaussian initialization of the policy and using gradient descent on the training cost, a generalization gap must necessarily exist.

The structure of this paper is outlined as follows: Section 2 discusses the motivation behind this work and the synthetic construction to abstract certain observation effects. Section 3 demonstrates numerous experiments using this synthetic construction that suggest implicit regularization is at work. Finally, Section 3.4 tests the implicit regularization hypothesis on CoinRun, as well as ablates various ImageNet architectures and margin metrics in the Appendix.

## 2 MOTIVATION AND RELATED WORK

We start by showing an example of observational overfitting in Figure 1. This example highlights the issues surrounding MDP's with rich, textured observations - specifically, the agent can use any features that are correlated with progress, even those which may not generalize across levels. This is an important issue for vision-based policies, as many times it is not obvious what part of the observation causes an agent to act or generalize.

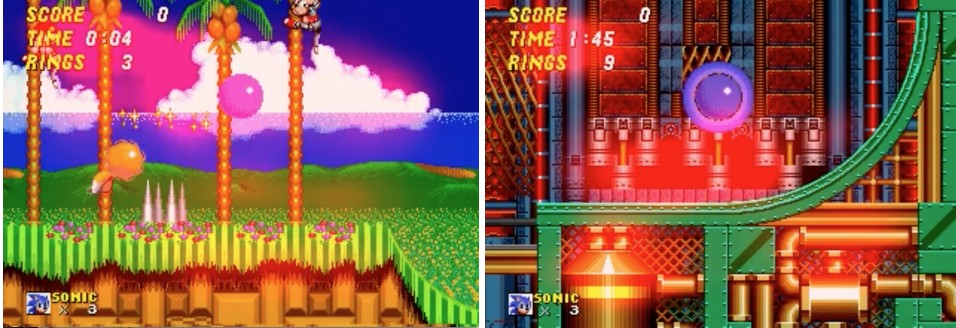

Figure 1: Example of observational overfitting in Sonic from Gym Retro (Nichol et al., 2018). Saliency maps (Greydanus et al., 2018) highlight (in red) the top-left timer and background objects such as clouds and textures because they are correlated with progress, as they move backwards while agent is moving forwards. The agent could memorize optimal actions for training levels even if its observation was *only from the timer*, and "blacking-out" the timer consistently improved generalization performance (see Appendix A.2.3).

Currently most architectures used in model-free RL are simple (with fewer than one million parameters) compared to the much larger and more complex ImageNet architectures used for classification.

This is due to the fact that most RL environments studied either have relatively simple and highly structured images (e.g. Atari) compared to real world images, or conveniently do not directly force the agent to observe highly detailed images. For instance in large scale RL such as DOTA2 (OpenAI, 2018) or Starcraft 2 (Vinyals et al., 2017), the agent observations are internal minimaps pertaining to object xy-locations, rather than human-rendered observations.

## 2.1 WHAT HAPPENS IN OBSERVATION SPACE?

Several artificial benchmarks (Zhang et al., 2018b; Gamrian & Goldberg, 2019) have been proposed before to portray this notion of overfitting, where an agent must deal with a changing background - however, a key difference in our work is that we explicitly require the "background" to be **correlated with the progress** rather than loosely correlated (e.g. through determinism between the background and the game avatar) or not at all. This makes a more explicit connection to *causal inference* (Arjovsky et al., 2019; Heinze-Deml & Meinshausen, 2019; Heinze-Deml et al., 2019) where *spurious correlations* between ungeneralizable features and progress may make training easy, but are detrimental to test performance because they induce false attributions.

Previously, many works interpret the decision-making of an agent through saliency and other network visualizations (Greydanus et al., 2018; Such et al., 2018) on common benchmarks such as Atari. Other recent works such as (Igl et al., 2019) analyze the interactions between noise-injecting explicit regularizations and the information bottleneck. However, our work is motivated by learning theoretic frameworks to capture this phenomena, as there is vast literature on understanding the generalization properties of SL classifiers (Vapnik & Chervonenkis, 1971; McAllester, 1999; Bartlett & Mendelson, 2002) and in particular neural networks (Neyshabur et al., 2015b; Dziugaite & Roy, 2017; Neyshabur et al., 2017; Bartlett et al., 2017; Arora et al., 2018c). For an RL policy with high-dimensional observations, we hypothesize its overfitting can come from more theoretically principled reasons, as opposed to purely good inductive biases on game images.

As an example of what may happen in high dimensional observation space, consider linear least squares regression task where given the set $X \in \mathbb{R}^{m \times d}$ and $Y \in \mathbb{R}^m$, we want to find $w \in \mathbb{R}^d$ that minimizes $\ell_{X,Y}(w) = \|Y - Xw\|^2$ where $m$ is the number of samples and $d$ is the input dimension. We know that if $X^\top X$ is full rank (hence $d \leq m$), $\ell_{X,Y}(.)$ has a unique global minimum $w^* = (X^\top X)^{-1}X^\top Y$. On the other hand if $X^\top X$ is not full rank (eg. when $m < d$), then there are many global minima $w^*$ such that $Y = Xw^*$ [1]. Luckily, if we use any gradient based optimization to minimize the loss and initialize with $w = 0$, the solution will only span column spaces of $X$ and converges to minimum $\ell_2$ norm solution among all global minima due to implicit regularization (Gunasekar et al., 2017). Thus a high dimensional observation space with a low dimensional state space can induce multiple solutions, some of which are not generalizable to other functions or MDP's but one could hope that implicit regularization would help avoiding this issue. We analyze this case in further detail for the convex one-step LQR case in Section 3.1 and Appendix A.4.3.

## 2.2 NOTATION

In the zero-shot framework for RL generalization, we assume there exists a distribution $\mathcal{D}$ over MDP's $\mathcal{M}$ for which there exists a fixed policy $\pi^{opt}$ that can achieve maximal return on expectation over MDP's generated from the distribution. An appropriate finite training set $\widehat{\mathcal{M}}_{train} = \{\mathcal{M}_1, \ldots, \mathcal{M}_n\}$ can then be created by repeatedly randomly sampling $\mathcal{M} \sim \mathcal{D}$. Thus for a MDP $\mathcal{M}$ and any policy $\pi$, expected episodic reward is defined as $R_{\mathcal{M}}(\pi)$.

In many empirical cases, the support of the distribution $\mathcal{D}$ is made by parametrized MDP's where some process, given a parameter $\theta$, creates a mapping $\theta \to \mathcal{M}_\theta$ (e.g. through procedural generation), and thus we may simplify notation and instead define a distribution $\Theta$ that induces $\mathcal{D}$, which implies a set of samples $\widehat{\Theta}_{train} = \{\theta_1, \ldots, \theta_n\}$ also induces a $\widehat{\mathcal{M}}_{train} = \{\mathcal{M}_1, \ldots, \mathcal{M}_n\}$, and we may redefine reward as $R_{\mathcal{M}_\theta}(\pi) = R_\theta(\pi)$.

---

[1] Given any $X$ with full rank $X^\top X$, it is possible to create many global minima by projecting the data onto high dimensions using a semi-orthogonal matrix $Z \in \mathbb{R}^{d \times d'}$ where $d' > m \geq d$ and $ZZ^\top = I_d$. Therefore, we the loss $\ell_{XZ,Y}(w) = \|Y - XZw\|^2$ will have many global optima $w^*$ with $Y = XZw^*$.

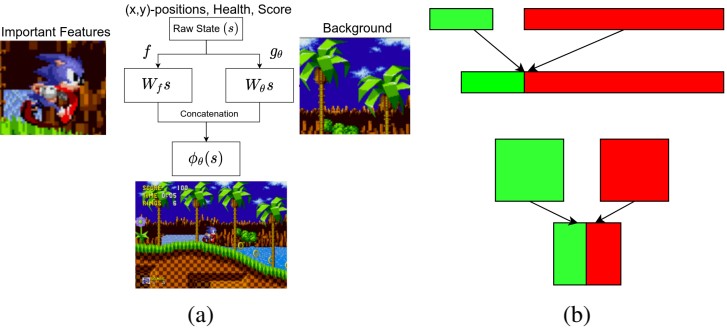

Figure 2: (a) Visual Analogy of the Observation Function. (b) Our combinations for 1-D (top) and 2-D (bottom) images for synthetic tasks.

As a simplified model of the observational problem from Sonic, we can construct a mapping $\theta \to \mathcal{M}_\theta$ by first fixing a base MDP $\mathcal{M} = (\mathcal{S}, \mathcal{A}, r, \mathcal{T})$, which corresponds to state space, action space, reward, and transition. The only effect of $\theta$ is to introduce an additional *observation function* $\phi_\theta : \mathcal{S} \to \mathcal{O}$, where the agent receives input from the high dimensional observation space $\mathcal{O}$ rather than from the state space $\mathcal{S}$. Thus, for our setting, $\theta$ actually parameterizes a POMDP family which can be thought of as simply a combination of a base MDP $\mathcal{M}$ and an observational function $\phi_\theta$, hence $\mathcal{M}_\theta = (\mathcal{M}, \phi_\theta)$.

Let $\widehat{\Theta}_{train} = \{\theta_1, \ldots, \theta_n\}$ be a set of $n$ i.i.d. samples from $\Theta$, and suppose we train $\pi$ to optimize reward against $\{\mathcal{M}_\theta : \theta \sim \widehat{\Theta}_{train}\}$. The objective $J_{\widehat{\Theta}}(\pi) = \frac{1}{|\widehat{\Theta}_{train}|} \sum_{\theta_i \in \widehat{\Theta}_{train}} R_{\theta_i}(\pi)$ is the average reward over this empirical sample. We want to generalize to the distribution $\Theta$, which can be expressed as the average episode reward $R$ over the full distribution, i.e. $J_\Theta(\pi) = \mathbb{E}_{\theta \sim \Theta}[R_\theta(\pi)]$. Thus we define the generalization gap as $J_{\widehat{\Theta}}(\pi) - J_\Theta(\pi)$.

## 2.3 SETUP

We can model the effects of Figure 1 more generally, not specific to sidescroller games. We assume that there is an underlying *state* $s$ (e.g. xy-locations of objects in a game), whose features may be very well structured, but that this state has been projected to a high dimensional observation space by $\phi_\theta$. To abstract the notion of generalizable and non-generalizable features, we construct a simple and natural candidate class of functions, where

$$\phi_\theta(s) = h(f(s), g_\theta(s)) \tag{1}$$

In this setup, $f(\cdot)$ is a function invariant for the entire MDP population $\Theta$, while $g_\theta(\cdot)$ is a function dependent on the sampled parameter $\theta$. $h$ is a "combination" function which combines the two outputs of $f$ and $g$ to produce a final observation. While $f$ projects this latent data into salient and important, *invariant* features such as the avatar, monsters, and items, $g_\theta$ projects the latent data to unimportant features that do not contribute to extra generalizable information, and can cause overfitting, such as the changing background or textures. A visual representation is shown in Figure 2. This is a simplified but still insightful model relevant in more realistic settings. For instance, in settings where $g_\theta$ does matter, learning this separation and task-identification (Yu et al., 2017; Peng et al., 2018) could potentially help fast adaptation in meta-learning (Finn et al., 2017). From now on, we denote this setup as the $(f, g)$-*scheme*.

This setting also leads to more interpretable generalization bounds - Lemma 2 of (Wang et al., 2019) provides a high probability $(1 - \delta)$ bound for the "intrinsic" generalization gap when $m$ levels are sampled: $gap \leq Rad_m(R_\Pi) + \mathcal{O}\left(\sqrt{\frac{\log(1/\delta)}{m}}\right)$, where

$$Rad_m(R_\Pi) = \mathbb{E}_{(\theta_1, \ldots, \theta_m) \sim \Theta^m}\left[\mathbb{E}_{\sigma \in \{-1, +1\}}\left[\sup_{\pi \in \Pi} \frac{1}{m} \sum_{i=1}^{m} \sigma_i R_{\theta_i}(\pi)\right]\right] \tag{2}$$

is the Rademacher Complexity under the MDP, where $\theta_i$ are the $\zeta_i$ parameters used in the original work, and the transition $\mathcal{T}$ and initialization $\mathcal{I}$ are fixed, therefore omitted, to accommodate our setting.

The Rademacher Complexity term captures how invariant policies in the set $\Pi$ with respect to $\theta$. For most RL benchmarks, this is not interpretable due to multiple confounding factors such as the varying level dynamics. For instance, it is difficult to imagine what behaviors or network weights a policy would possess in order to produce the same total rewards, *regardless of changing dynamics.*

However, in our case, because the environment parameters $\theta$ are only from $g_\theta$, the Rademacher Complexity is directly based on how much the policy "looks at" $g_\theta$. More formally, let $\Pi^*$ be the set of policies $\pi^*$ which are not be affected by changes in $g_\theta$; i.e. $\nabla_\theta \pi^*(\phi_\theta(s)) = 0 \ \forall s$ and thus $R_\theta(\pi^*) = R_{const} \ \forall \theta$, which implies that the environment parameter $\theta$ has no effect on the reward; hence $Rad_m(R_{\Pi^*}) = \mathbb{E}_{\sigma \in \{-1,+1\}} \left[ \sup_{\pi^* \in \Pi^*} \frac{1}{m} \sum_{i=1}^m \sigma_i R_{const} \right] = 0$.

### 2.4 ARCHITECTURE AND IMPLICIT REGULARIZATION

Normally in a MDP such as a game, the concatenation operation may be dependent on time (e.g. textures move around in the frame). In the scope of this work, we simplify the concatenation effect and assume $h(\cdot)$ is a static concatenation, but still are able to demonstrate insightful properties. We note that this inductive bias on $h$ allows *explicit regularization* to trivially solve this problem, by penalizing a policy's first layer that is used to "view" $g_\theta(s)$ (Appendix A.1.1), hence we only focus on implicit regularizations.

This setting is naturally attractive to analyzing architectural differences, as it is more closely related in spirit to image classifiers and SL. One particular line of work to explain the effects of certain architectural modifications in SL such as overparametrization and residual connections is *implicit regularization* (Neyshabur et al., 2015a; Gunasekar et al., 2017; Neyshabur, 2017), as overparametrization through more layer depth and wider layers has proven to have no $\ell_p$-regularization equivalent (Arora et al., 2019), but rather precondition the dynamics during training. Thus, in order to fairly experimentally measure this effect, we always use fixed hyperparameters and only vary based on architecture. In this work, we only refer to *architectural* implicit regularization techniques, which do not have a explicit regularization equivalent. Some techniques e.g. coordinate descent (Bradley et al., 2011) are equivalent to explicit $\ell_1$-regularization.

## 3 EXPERIMENTS

### 3.1 OVERPARAMTERIZED LQR

We first analyze the case of the LQR as a surrogate for what may occur in deep RL, which has been done before for various topics such as sample complexity (Dean et al., 2019) and model-based RL (Tu & Recht, 2019). This is analogous to analyzing linear/logistic regression (Kakade et al., 2008; McAllester, 2003) as a surrogate to understanding extensions to deep SL techniques (Neyshabur et al., 2018a; Bartlett et al., 2017). In particular, this has numerous benefits - the cost (negative of reward) function is deterministic, and allows exact gradient descent (i.e. the policy can differentiate through the cost function) as opposed to necessarily using stochastic gradients in normal RL, and thus can cleanly provide evidence of implicit regularization. Furthermore, in terms of gradient dynamics and optimization, LQR readily possesses nontrivial qualities compared to linear regression, as the LQR cost is a non-convex function but all of its minima are *global minima* (Fazel et al., 2018).

To show that overparametrization alone is an important implicit regularizer in RL, LQR allows the use of linear policies (and consequently also allows stacking linear layers) without requiring a stochastic output such as discrete Gumbel-softmax or for the continuous case, a parametrized Gaussian. This is setting able to show that overparametrization alone can affect gradient dynamics, and is not a consequence of extra representation power due to additional non-linearities in the policy. There have been multiple recent works on this linear-layer stacking in SL and other theoretical problems such as matrix factorization and matrix completion (Arora et al., 2018b;a; Gunasekar et al., 2017), but to our knowledge, we are the first to analyze this case in the context of RL generalization.

We explicitly describe setup as follows: for a given $\theta$, we let $f(s) = W_c \cdot s$, while $g_\theta(s) = W_\theta \cdot s$ where $W_c, W_\theta$ are semi-orthogonal matrices, to prevent information loss relevant to outputting the optimal action, as the state is transformed into the observation. Hence, if $s_t$ is the underlying state at time $t$, then the observation is $o_t = \begin{bmatrix} W_c \\ W_\theta \end{bmatrix} s_t$ and thus the action is $a_t = K o_t$, where $K$ is the policy matrix. While $W_c$ remains a constant matrix, we sample $W_\theta$ randomly, using the "level ID" integer $\theta$ as the seed for random generation. In terms of dimensions, if $s$ is of shape $d_{state}$, then $f$ also projects to a shape of $d_{state}$, while $g_\theta$ projects to a much larger shape $d_{noise}$, implying that the observation to the agent is of dimension $d_{signal} + d_{noise}$. In our experiments, we set as default $(d_{signal}, d_{noise}) = (100, 1000)$.

If $P_\star$ is the unique minimizer of the original cost function, then the unique minimizer of the population cost is $K_\star = \begin{bmatrix} W_c P_\star^{\mathsf{T}} \\ 0 \end{bmatrix}^{\mathsf{T}}$. However, if we have a single level, then there exist multiple solutions, for instance $\begin{bmatrix} \alpha W_c P_\star^{\mathsf{T}} \\ (1-\alpha) W_\theta P_\star^{\mathsf{T}} \end{bmatrix}^{\mathsf{T}}$ $\forall \alpha$. This extra bottom component $W_\theta P_\star^{\mathsf{T}}$ causes overfitting. In Appendix A.4.3, we show that in the 1-step LQR case (which can be extended to convex losses whose gradients are linear in the input), gradient descent cannot remove this component, and thus overfitting necessarily occurs.

Furthermore, we find that increasing $d_{noise}$ increases the generalization gap in the LQR setting. This is empirically verified in Figure 3 using an actual non-convex LQR loss, and the results suggest that the gap scales by $\mathcal{O}(\sqrt{d_{noise}})$. In terms of overparametrization, we experimentally added more $(100 \times 100)$ linear layers $K = K_0 K_1, ..., K_j$ and increased widths for a 2-layer case (Figure 3), and observe that both settings reduce the generalization gap, and also reduce the norms (spectral, nuclear, Frobenius) of the final end-to-end policy $K$, without changing its expressiveness. This suggests that gradient descent under overparametrization implicitly biases the policy towards a "simpler" model in the LQR case.

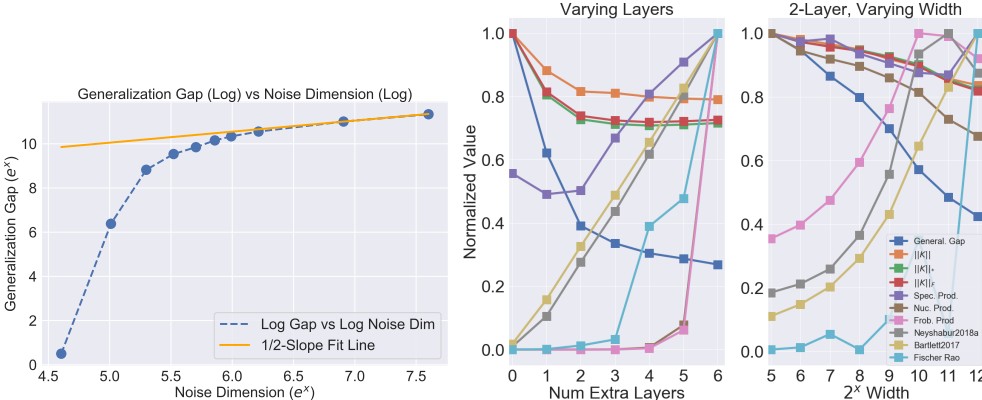

Figure 3: (**Left**) We show that the generalization gap vs noise dimension is tight as the noise dimension increases, showing that this bound is accurate. (**Middle** and **Right**) LQR Generalization Gap vs Number of Intermediate Layers. We plotted different $\Phi = \sum_{i=0}^{j} \frac{\|A\|_*}{\|A\|}$ terms without exponents, as powers of those terms are monotonic transforms since $\frac{\|A\|_*}{\|A\|} \geq 1$ $\forall A$ and $\|A\|_* = \|A\|_F, \|A\|_1$. We see that the naive spectral bound diverges at 2 layers, and the weight-counting sums are too loose.

As a surrogate model for deep RL, one may ask if the generalization gap of the final end-to-end policy $K$ can be predicted by functions of the layers $K_0, ..., K_j$. This is an important question as it is a required base case for predicting generalization when using stochastic policy gradient with nonlinear activations such as ReLU or Tanh. From examining the distribution of singular values on $K$ (Appendix A.1.1), we find that more layers does not bias the policy towards a low rank solution in the nonconvex LQR case, unlike (Arora et al., 2018b) which shows this does occur for matrix completion, and in general, convex losses. Ultimately, we answer in the negative: intriguingly, SL bounds have very little predictive power in the RL domain case.

To understand why SL bounds may be candidates for the LQR case, we note that as a basic smoothness bound $C(K) - C(K') \leq \mathcal{O}(\|K - K'\|)$ (Appendix A.4) can lead to very similar reasoning with SL bounds. Since our setup is similar to SL in that "LQR levels" which may be interpreted as a dataset, we use bounds of the form $\Delta \cdot \Phi$, where $\Delta$ is a "macro" product term $\Delta = \prod_{i=0}^{j} \|K_i\| \geq \left\|\prod_{i=0}^{j} K_i\right\|$ derivable from the fact that $\|AB\| \leq \|A\| \|B\|$ in the linear case, and $\Phi$ is a *weight-counting* term which deals with the overparametrized case, such as $\Phi = \sum_{i=0}^{j} \frac{\|K_i\|_F^2}{\|K_i\|^2}$ (Neyshabur et al., 2018a) or $\Phi = \left(\sum_{i=0}^{j} \left(\frac{\|K_i\|_1}{\|K_i\|}\right)^{2/3}\right)^3$ (Bartlett et al., 2017). However, the $\Phi$ terms increase too rapidly as shown in Figure 3. Terms such as Frobenius product (Golowich et al., 2018) and Fischer-Rao (Liang et al., 2019) are effective for the SL depth case, but are both ineffective in the LQR depth case. For width, the only product which is effective is the nuclear norm product.

## 3.2 Projected Gym Environments

In Section 3.1, we find that observational overfitting exists and overparametrization potentially helps in the linear setting. In order to analyze the case when the underlying dynamics are nonlinear, we let $\mathcal{M}$ be a classic Gym environment and we generate a $\mathcal{M}_\theta = (\mathcal{M}, w_\theta)$ by performing the exact same $(f, g)$-scheme as the LQR case, i.e. sampling $\theta$ to produce an observation function $w_\theta(s) = \begin{bmatrix} W_c \\ W_\theta \end{bmatrix} s$. We again can produce training/test sets of MDPs by repeatedly sampling $\theta$, and for policy optimization, we use Proximal Policy Gradient (Schulman et al., 2017).

Although bounds on the smoothness term $R_\theta(\pi) - R_\theta(\pi')$ affects upper bounds on Rademacher Complexity (and thus generalization bounds), we have no such theoretical guarantees in the Mujoco case as it is difficult to analyze the smoothness term for complicated transitions such as Mujoco's physics simulator. However, in Figure 4, we can observe empirically that the underlying state dynamics has a significant effect on generalization performance as the policy nontrivially increased test performance such as in CartPole-v1 and Swimmer-v2, while it could not for others. This suggests that the Rademacher complexity and smoothness on the reward function vary highly for different environments.

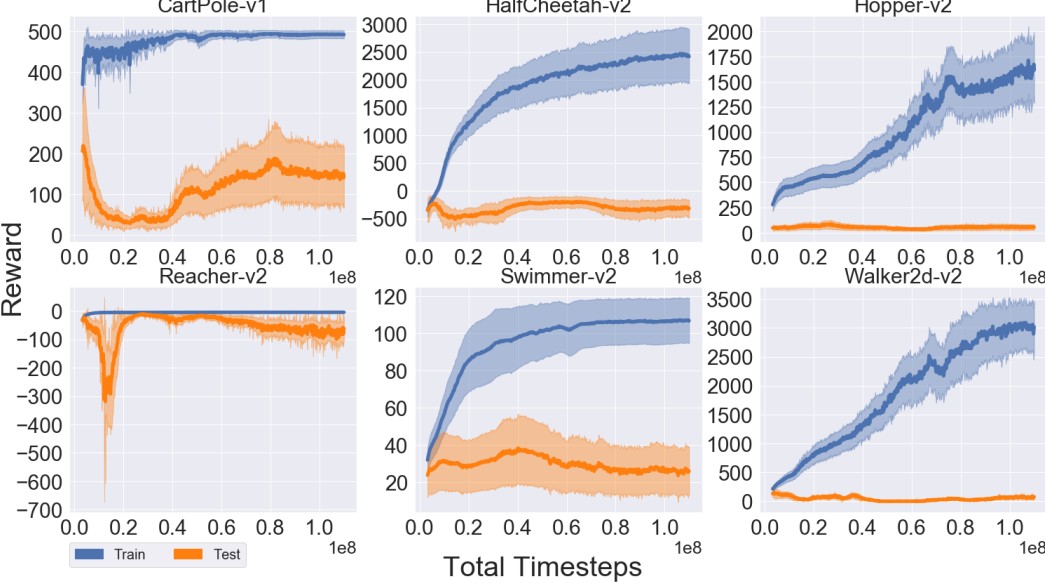

Figure 4: Each Mujoco task is given 10 training levels (randomly sampling $g_\theta$ parameters). We used a 2-layer ReLU policy, with 128 hidden units each. Dimensions of outputs of $(f, g)$ were $(30, 100)$ respectively.

Even though it is common practice to use basic (2-layer) MLPs in these classic benchmarks, there are highly nontrivial generalization effects from modifying on this class of architectures. Our results in

Figures 5 and 6 show that increasing width and depth for basic MLPs can increase generalization and is significantly dependent on the choice of activation, and other implicit regularizations such as using residual layers can also improve generalization. Specifically, switching between ReLU and Tanh activations produces different results during overparametrization. For instance, increasing Tanh layers improves generalization on CartPole-v1, and width increase with ReLU helps on Swimmer-v2. Tanh is noted to consistently improve generalization performance. However, stacking Tanh layers comes at a cost of also producing vanishing gradients which can produce subpar training performance, for e.g. HalfCheetah. To allow larger depths, we use ReLU *residual* layers, which also improves generalization and stabilizes training.

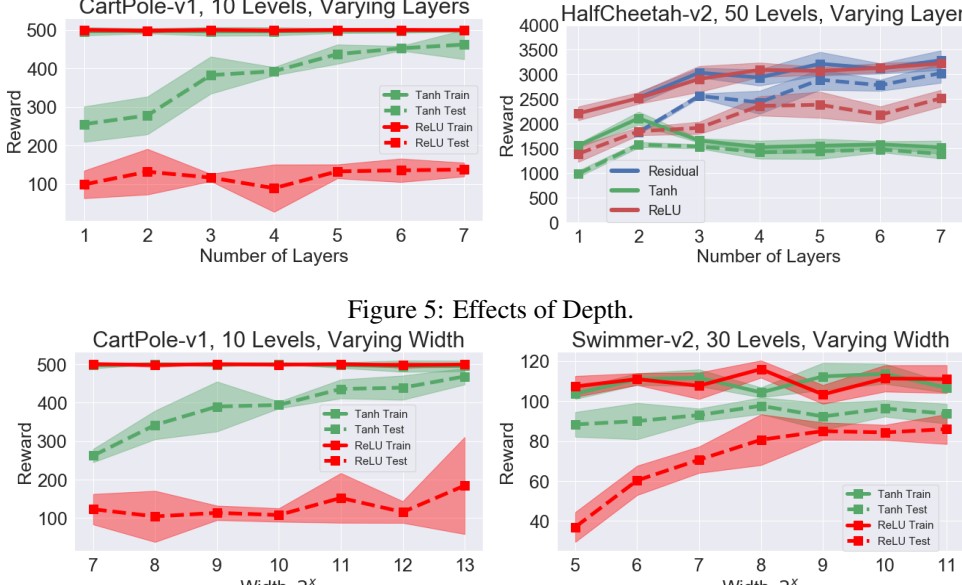

Figure 5: Effects of Depth.

Figure 6: Effects of Width.

Previous work (Zhang et al., 2018c) did not find such an architectural pattern for GridWorld environments, suggesting that this effect may exist primarily for observational overfitting cases. While there have been numerous works which avoid overparametrization on simplifying policies (Rajeswaran et al., 2017; Mania et al., 2018) or compactifying networks (Choromanski et al., 2018; Gaier & Ha, 2019), we instead find that there are generalization benefits to overparametrization even in the nonlinear control case.

### 3.3 DECONVOLUTIONAL PROJECTIONS

From the above results with MLPs, one may wonder if similar results may carry to convolutional networks, as they are widely used for vision-based RL tasks. As a ground truth reference for our experiment, we the canonical networks proven to generalize well in the dataset CoinRun, which are from worst to best, NatureCNN Mnih et al. (2013), IMPALA Espeholt et al. (2018), and IMPALA-LARGE (IMPALA with more residual blocks and higher convolution depths), which have respective parameter numbers (600K, 622K, 823K).

We setup a similar $(f, g)$-scheme appropriate for the inductive bias of convolutions, by passing the vanilla Gym 1D state corresponding to joint locations and velocities, through multiple deconvolutions. We do so rather than using the RGB image from `env.render()` to enforce that the actual state is indeed low dimensional and minimize complications in experimentation, as e.g. inference of velocity information would require frame-stacking.

Specifically in our setup, we project the actual state to a fixed length, reshaping it into a square, and replacing $f$ and $g_\theta$ both with the same orthogonally-initialized *deconvolution* architecture to each produce a $84 \times 84$ image (but $g_\theta$'s network weights are still generated by $\theta_1, ..., \theta_m$ similar to before). We combine the two outputs by using one half of the "image" from $f$, and one half from $g_\theta$, as shown back in Figure 2.

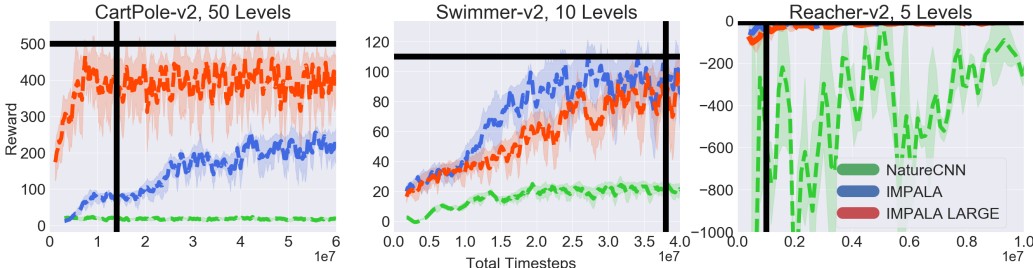

Figure 7: Performance of architectures in the synthetic Gym-Deconv dataset. To cleanly depict test performance, training curves are replaced with horizontal (max env. reward) and vertical black lines (avg. timestep when all networks reach max reward).

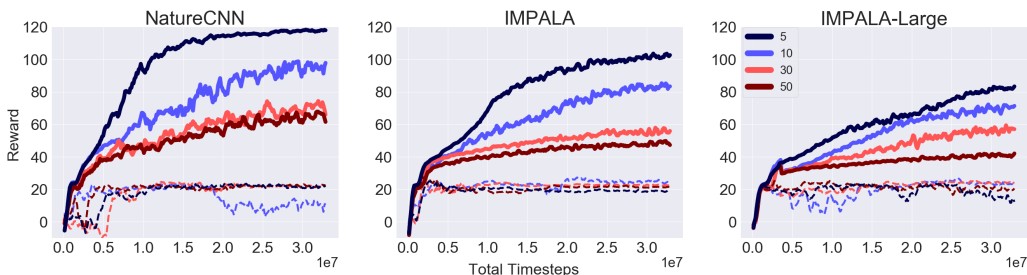

Figure 8: We only show the observation from $g_\theta(s)$, which tests memorization capacity on Swimmer-v2.

Figure 7 shows that the same ranking between the three architectures exists as well on the Gym-Deconv dataset. We show that generalization ranking among NatureCNN/IMPALA/IMPALA-LARGE remains the same regardless of whether we use our synthetic constructions or CoinRun. This suggests that the RL generalization quality of a convolutional architecture is not limited to *real world data*, as our test purely uses numeric observations, which are not based on a human-prior. From these findings, one may conjecture that these RL generalization performances are highly correlated and may be due to common factors.

One of these factors we suggest is due to implicit regularization. In order to support this claim, we perform a memorization test by only showing $g_\theta$'s output to the policy. This makes the dataset impossible to generalize to, as the policy network cannot invert every single observation function $\{g_{\theta_1}(\cdot), g_{\theta_2}(\cdot), ..., g_{\theta_n}(\cdot)\}$ simultaneously. Zhang et al. (2018c) also constructs a memorization test for mazes and grid-worlds, and showed that more parameters increased the memorization ability of the policy. While it is intuitive that more parameters would incur more memorization, we show in Figure 8 that this is perhaps not a complete picture when implicit regularization is involved.

Using the underlying MDP as a Swimmer-v2 environment, we see that NatureCNN, IMPALA, IMPALA-LARGE have reduced memorization performances. IMPALA-LARGE, which has more depth parameters and more residual layers (and thus technically has more capacity), memorizes *less* than IMPALA due its inherent inductive bias. While memorization performance is dampened in 8, we perform another deconvolution memorization test using an LQR as the underlying MDP in Appendix A.1.1 that shows that there can exist specific hard limits to memorization, which also follows the same ranking above.

## 3.4 OVERPARAMETRIZATION IN COINRUN

We further test our overparametrization hypothesis from Sections 3.1, 3.2 to the CoinRun benchmark, using unlimited levels for training. For MLP networks, we downsized CoinRun from native $64 \times 64$ to $32 \times 32$, and flattened the $32 \times 32 \times 3$ image for input to an MLP. Two significant differences from the synthetic cases are that 1. Inherent dynamics are changing per level in CoinRun, and 2. The relevant and irrelevant CoinRun features change locations across the 1-D input vector. Regardless, in Figure 9, we show that overparametrization can still improve generalization in this more realistic

RL benchmark, much akin to (Neyshabur et al., 2018b) which showed that overparametrization for MLP's improved generalization on $32 \times 32 \times 3$ CIFAR-10.

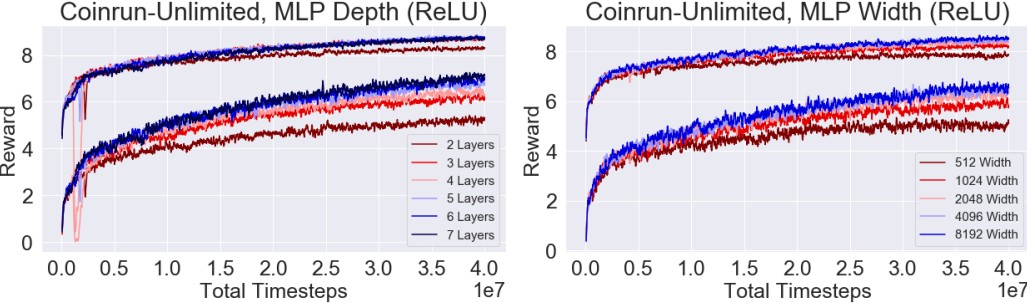

Figure 9: Overparametrization improves generalization for CoinRun.

While we also extend the case of large-parameter convolutional networks using ImageNet networks in Appendix A.2.1, an important question is how to predict the generalization gap only from the training phase. A particular set of metrics, popular in the SL community are *margin distributions* (Jiang et al., 2018; Bartlett et al., 2017), as they deal with the case for softmax outputs which do not explicitly penalize the weight norm of a network, by normalizing the "confidence" margin of the logit outputs. While using margins on state-action pairs (from an on-policy replay buffer) is not technically rigorous, one may be curious to see if they have predictive power, especially as MLPs are relatively simple to norm-bound. We plotted these margin distributions in Appendix A.2.2, but found that the weight norm bounds used in SL are simply too dominant for this RL case. This, with the bound results found earlier for the LQR case, suggests that current norm bounds are simply too loose for the RL case even though we have shown overparametrization helps generalization in RL, and hopefully this motivates more of the study of such theory.

## 4 CONCLUSION

We have identified and isolated a key component of overfitting in RL as the particular case of "observational overfitting", which is particularly attractive for studying architectural implicit regularizations. We have analyzed this setting extensively, by examining 3 main components:

1. The analytical case of LQR and linear policies under exact gradient descent, which lays the foundation for understanding theoretical properties of networks in RL generalization.
2. The empirical but principled Projected-Gym case for both MLP and convolutional networks which demonstrates the effects of neural network policies under nonlinear environments.
3. The large scale case for CoinRun, which can be interpreted as a case where relevant features are moving across the input, where empirically, MLP overparametrization also improves generalization.

We noted that current network policy bounds using ideas from SL are unable to explain overparametrization effects in RL, which is an important further direction. In some sense, this area of RL generalization is an extension of static SL classification from adding extra RL components. For instance, adding a nontrivial "combination function" between $f$ and $g_\theta$ that is dependent on time (to simulate how object pixels move in a real game) is both an RL generalization issue and potentially video classification issue, and extending results to the memory-based RNN case will also be highly beneficial.

Furthermore, it is unclear whether such overparametrization effects would occur in off-policy methods such as Q-learning and also ES-based methods. In terms of architectural design, recent works (Jacot et al., 2018; Garriga-Alonso et al., 2019; Lee et al., 2019) have shed light on the properties of asymptotically overparametrized neural networks in the infinite width and depth cases and their performance in SL. Potentially such architectures (and a corresponding training algorithm) may be used in the RL setting which can possibly provide benefits, one of which is generalization as shown in this paper. We believe that this work provides an important initial step towards solving these future problems.

ACKNOWLEDGEMENTS

We would like to thank John Schulman for very helpful guidance over the course of this work. We also wish to thank Chiyuan Zhang, Ofir Nachum, Aurick Zhou, Daniel Seita, Alexander Irpan, and the OpenAI team for fruitful comments and discussions during the course of this work.

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

## A.1 FULL PLOTS FOR LQR AND FG-GYM

### A.1.1 LQR

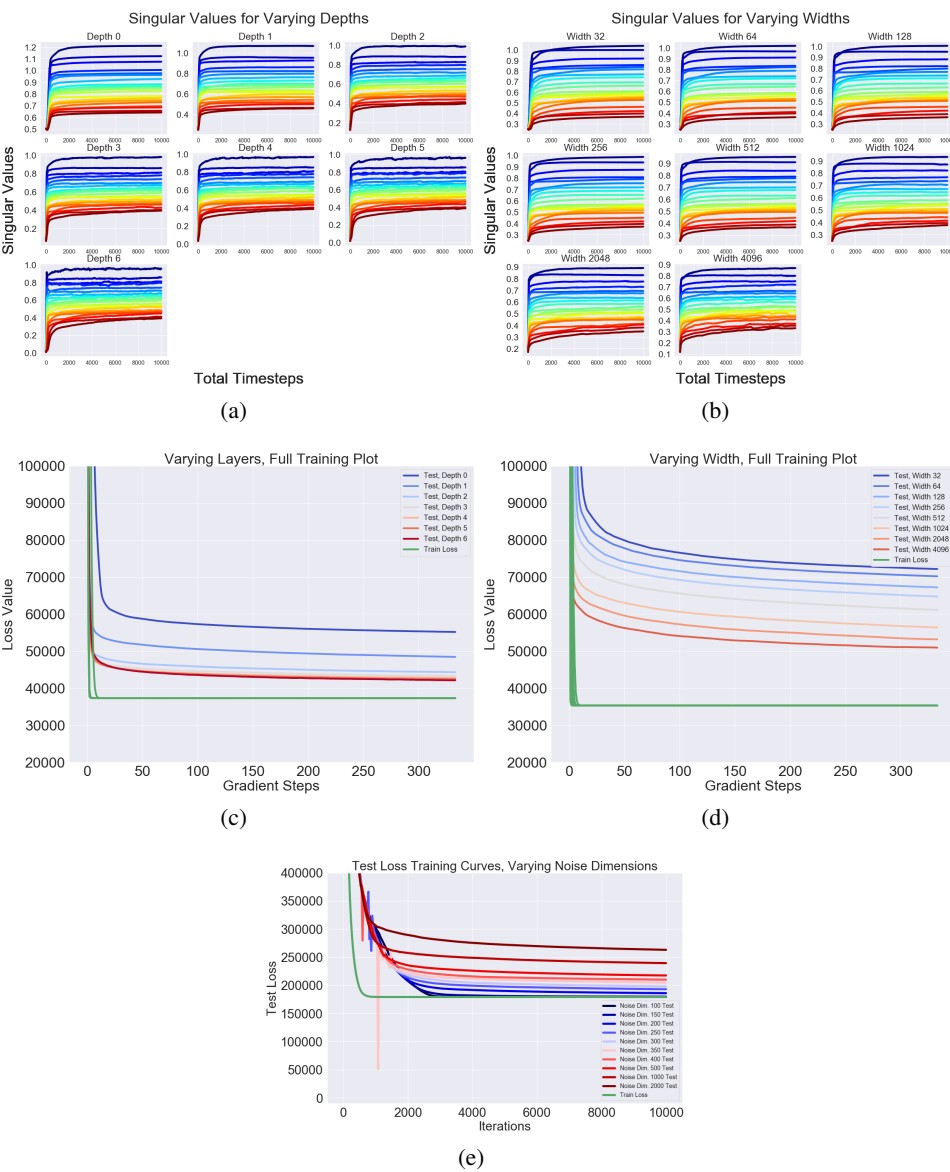

Figure A1: (a,b): Singular Values for varying depths and widths. (c,d): Train and Test Loss for varying widths and depths. (e): Train and Test Loss for varying Noise Dimensions.

We further verify that *explicit regularization* (norm based penalties) also reduces generalization gaps. However, explicit regularization may be explained due to the bias of the synthetic tasks, since the first layer's matrix may be regularized to only "view" the output of $f$, especially as regularizing the first layer's weights substantially improves generalization.

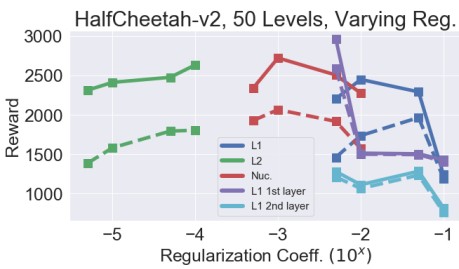

Figure A2: Explicit Regularization on layer norms.

We provide another deconvolution memorization test, using an LQR as the underlying MDP. While fg-Gym-Deconv shows that memorization performance is dampened, this test shows that there can exist specific hard limits to memorization. Specifically, NatureCNN can memorize 30 levels, but not 50; IMPALA can memorize 2 levels but not 5; IMPALA-LARGE cannot memorize 2 levels at all.

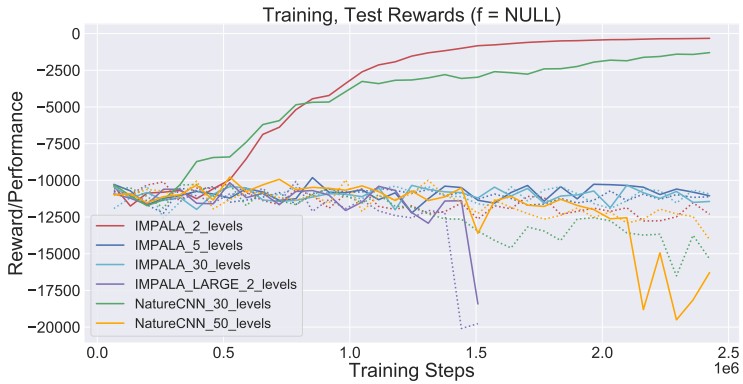

Figure A3: Deconvolution memorization test using LQR as underlying MDP.

## A.2 EXTENDED LARGE RL RESULTS

### A.2.1 LARGE IMAGENET MODELS FOR COINRUN

For reference, we also extend the case of large-parameter convolutional networks using ImageNet networks. We experimentally verify in Table 1 that large ImageNet models perform very differently in RL than SL. We note that default network with the highest test reward was IMPALA-LARGE-BN (IMPALA-LARGE, with Batchnorm) at $\approx 5.5$ test score.

In order to verify that this is inherently a *feature learning* problem rather than a *combinatorial problem* involving objects, such as in (Santoro et al., 2018), we show that state-of-the-art attention mechanisms for RL such as Relational Memory Core (RMC) using pure attention on raw $32 \times 32$ pixels does not perform well here, showing that a large portion of generalization and transfer must be based on correct convolutional setups.

| Architecture | Coinrun-100 (Train, Test) |
|---|---|
| AlexNet-v2 | (10.0, 3.0) |
| CifarNet | (10.0, 3.0) |
| IMPALA-LARGE-BN | (10.0, 5.5) |
| Inception-ResNet-v2 | (10.0, 6.5) |
| Inception-v4 | (10.0, 6.0) |
| MobileNet-v1 | (10.0, 5.5) |
| MobileNet-v2 | (10.0, 5.5) |
| NASNet-CIFAR | (10.0, 4.0) |
| NASNet-Mobile | (10.0, 4.5) |
| ResNet-v2-50 | (10.0, 5.5) |
| ResNet-v2-101 | (10.0, 5.0) |
| ResNet-v2-152 | (10.0, 5.5) |
| RMC32x32 | (9.0, 2.5) |
| ShakeShake | (10.0, 6.0) |
| VGG-A | (9.0, 3.0) |
| VGG-16 | (9.0, 3.0) |

Table 1: Raw Network Performance (rounded to nearest 0.5) on CoinRun, 100 levels. Images scaled to default image sizes ($32 \times 32$ or $224 \times 224$) depending on network input requirement. See Appendix A.2.1 for training curves.

We provide the training/testing curves for the ImageNet/large convolutional models used. Note the following:

1. RMC32x32 projects the native image from CoinRun from $64 \times 64$ to $32 \times 32$, and uses all pixels as components for attention, after adding the coordinate embedding found in (Santoro et al., 2018). Optimal parameters were (mem_slots = 4, head_size = 32, num_heads = 4, num_blocks = 2, gate_style = 'memory').
2. Auxiliary Loss in ShakeShake was not used during training, only the pure network.
3. VGG-A is a similar but slightly smaller version of VGG-16.

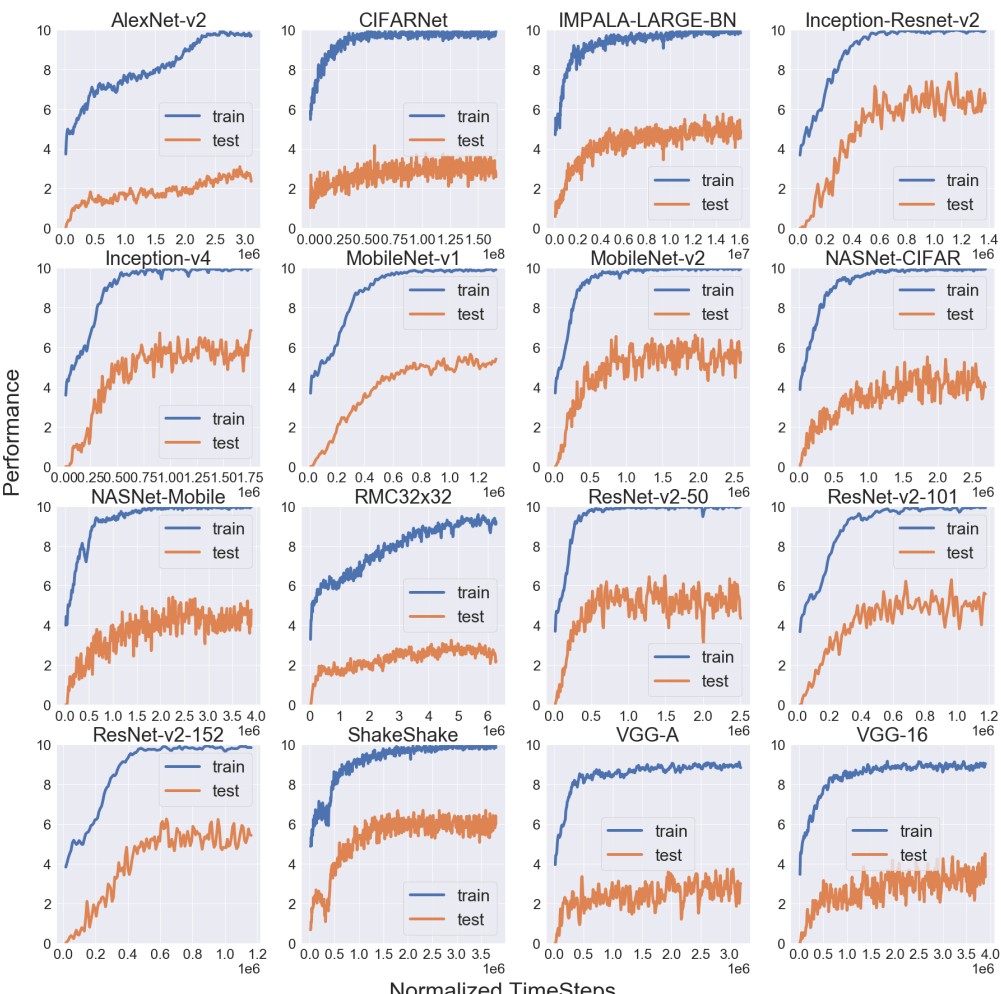

Figure A4: Large Architecture Training/Testing Curves (Smoothed).

### A.2.2 Do State-Action Margin Distributions Predict Generalization in RL?

A key question is how to predict the generalization gap only from the training phase. A particular set of metrics, popular in the SL community are *margin distributions* (Jiang et al., 2018; Bartlett et al., 2017), as they deal with the case for softmax categorical outputs which do not explicitly penalize the weight norm of a network, by normalizing the "confidence" margin of the logit outputs. While using margins on state-action pairs (from an on-policy replay buffer) is not technically rigorous, one may be curious to see if they have predictive power, especially as MLP's are relatively simple to norm-bound, and as seen from the LQR experiments, the norm of the policy may be correlated with the generalization performance.

For a policy, the the margin distribution will be defined as $(x, y) \rightarrow \frac{F_\pi(x)_y - \max_{i \neq y} F_\pi(x)_i}{\mathcal{R}_\pi \|S\|_2 / n}$, where $F_\pi(x)_y$ is the logit value (before applying softmax) of output $y$ given input $x$, and $S$ is the matrix of states in the replay buffer, and $\mathcal{R}_\pi$ is a norm-based Lipschitz measure on the policy network logits. In general, $\mathcal{R}_\pi$ is a bound on the Lipschitz constant of the network but can also be simply expressions which allow the margin distribution to have high correlation with the generalization gap. Thus, we use measures inspired by recent literature in SL in which we designate Spectral-L1, Distance, and Spectral-Frobenius measures for $\mathcal{R}_\pi$, and we replace the classical supervised learning pair $(x, y) = (s, a)$ with the state-action pairs found on-policy. [2]

The expressions for $\mathcal{R}_\pi$ (after removing irrelevant constants) are as follows, with their analogous papers:

1. Spectral-L1 measure: $\left(\prod_{i=1}^d \|W_i\|\right) \left(\sum_{i=1}^d \frac{\|W_i\|_1^{2/3}}{\|W_i\|^{2/3}}\right)^{3/2}$ (Bartlett et al., 2017)

2. Distance measure: $\sqrt{\sum_{i=1}^d \|W_i - W_i^0\|_F^2}$ (Nagarajan & Kolter, 2019)

3. Spectral-Fro measure: $\sqrt{\ln(d) \prod_{i=1}^d \|W_i\|^2 \sum_{j=1}^d \frac{\|W_j - W_j^0\|_F^2}{\|W_j\|^2}}$ (Neyshabur et al., 2018a)

We verify in Figure A5, that indeed, simply measuring the raw norms of the policy network is a poor way to predict generalization, as it generally increases even as training begins to plateau. This is inherently because the softmax on the logit output does not penalize arbitrarily high logit values, and hence proper normalization is needed.

The margin distribution converges to a fixed distribution even long after training has plateaued. However, unlike SL, the margin distribution is conceptually not fully correlated with RL generalization on the total reward, as a policy overconfident in some state-action pairs does not imply bad testing performance. This correlation is stronger if there are Lipschitz assumptions on state-action transitions, as noted in (Wang et al., 2019). For empirical datasets such as CoinRun, a metric-distance between transitioned states is ill-defined however. Nevertheless, the distribution over the on-policy replay buffer at each policy gradient iteration is a rough measure of overall confidence.

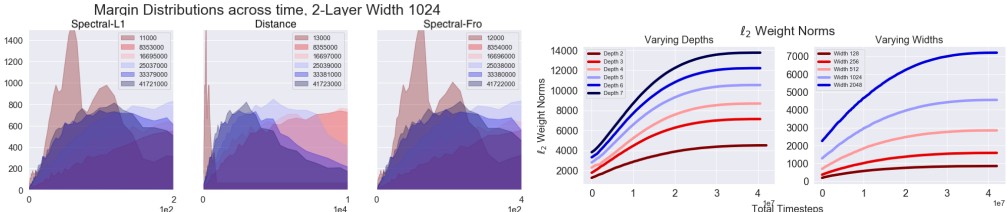

Figure A5: Margin Distributions across training.

We note that there are two forms of modifications, *network dependent* (explicit modifications to the policy - norm regularization, dropout, etc.) and *data dependent* (modifications only to the data in the replay buffer - action stochasticity, data augmentation, etc.). Ultimately however, we find that current

---

[2]We removed the training sample constant $m$ from all original measures as this is ill-defined for the RL case, when one can generate infinitely many $(s, a)$ pairs. Furthermore, we used the original $\|W_i\|_1$ in the numerator found in the first version of (Bartlett et al., 2017) rather than the current $\|W_i\|_{1,2}$.

norm measures $R_\pi$ become too dominant in the fraction, leading to the monotonic decreases in the means of the distributions as we increase parametrization.

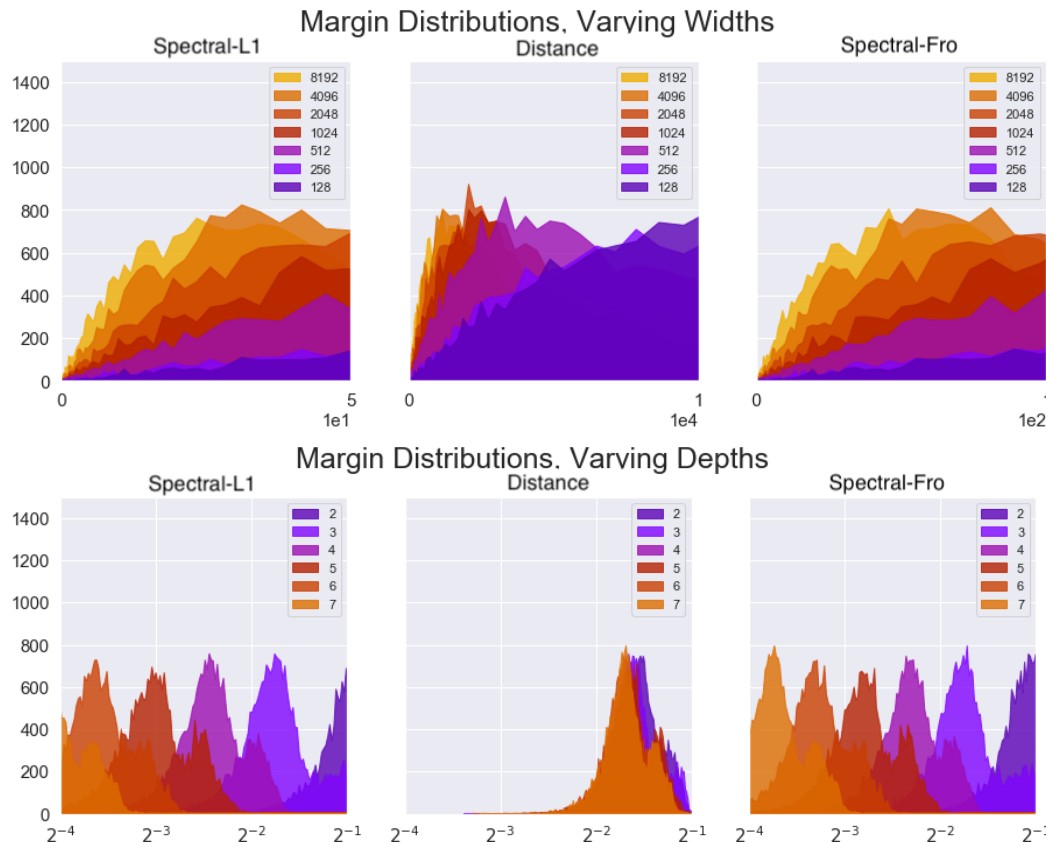

Figure A6: Margin Distributions at the end of training.

### A.2.3 GYM-RETRO (SONIC)

In the Gym-Retro benchmark using Sonic (Nichol et al., 2018), the agent is given 47 training levels with rewards corresponding to increases in horizontal location. The policy is trained until 5k reward. At test time, 11 unseen levels are partitioned into starting positions, and the rewards are measured and averaged.

We briefly mention that the agent strongly overfits to the scoreboard (i.e. an artifact correlated with progress in the level), which may be interpreted as part of the output of $g_\theta(\cdot)$. In fact, the agent is still able to train to 5k reward from purely observing the timer as the observation. By blacking out this scoreboard with a black rectangle, we see an increase in test performance.

| Settings | IMPALA | NatureCNN |
|---|---|---|
| Blackout | $1250 \pm 40$ | $1141 \pm 40$ |
| NoBlackout | $1130 \pm 40$ | $1052 \pm 40$ |

Table 2: IMPALA vs NatureCNN test rewards, with and without Blackout.

### A.3 HYPERPARAMETERS AND EXACT SETUPS

### A.3.1 EXACT INFINITE LQR

For infinite horizon case, see (Fazel et al., 2018) for the the full solution and notations. Using the same notation $(A, B, Q, R)$, denote $C(K) = \sum_{x_0 \sim \mathcal{D}} x_0^T P_K x_0$ as the cost and $u_t = -K x_t$ as the

policy, where $P_K$ satisfies the infinite case for the Lyapunov equation:

$$P_K = Q + K^T R K + (A - BK)^T P_K (A - BK) \tag{3}$$

We may calculate the precise LQR cost by vectorizing (i.e. flattening) both sides' matrices and using the Kroncker product $\otimes$, which leads to a linear regression problem on $P_K$, which has a precise solution, implementable in TensorFlow:

$$\text{vec}(P_K) = \text{vec}(Q) + \text{vec}(K^T R K) + \left[ (A - BK)^T \otimes (A - BK)^T \right] \text{vec}(P_K) \tag{4}$$

$$\left[ I_{n^2} - (A - BK)^T \otimes (A - BK)^T \right] \text{vec}(P_K) = \text{vec}(Q) + \text{vec}(K^T R K) \tag{5}$$

| Parameter | Generation |
|---|---|
| $A$ | Orthogonal initialization, scaled 0.99 |
| $B$ | $I_n$ |
| $Q$ | $I_n$ |
| $R$ | $I_n$ |
| $n$ | 10 |
| $K_i \quad \forall i$ | Orthogonal Initialization, scaled 0.5 |

Table 3: Hyperparameters for LQR

### A.3.2 PROJECTION METHOD

The basis for producing $f, g_\theta$ outputs is due to using batch matrix multiplication operations, or "BMV", where the same network architecture uses different network weights for each batch dimension, and thus each entry in a batchsize of $B$ will be processed by the same architecture, but with different network weights. This is to simulate the effect of $g_{\theta_i}$. The numeric ID $i$ of the environment is used as an index to collect a specific set of network weights $\theta_i$ from a global memory of network weights (e.g. using `tensorflow.gather`). We did not use nonlinear activations for the BMV architectures, as they did not change the outcome of the results.

| Architecture | Setup |
|---|---|
| BMV-Deconv | (filtersize = 2, stride = 1, outchannel = 8, padding = "VALID") |
| | (filtersize = 4, stride = 2, outchannel = 4, padding = "VALID") |
| | (filtersize = 8, stride = 2, outchannel = 4, padding = "VALID") |
| | (filtersize = 8, stride = 3, outchannel = 3, padding = "VALID") |
| BMV-Dense | $f$ : Dense 30, $g$ : Dense 100 |

### A.3.3 IMAGENET MODELS

For the networks used in the supervised learning tasks, we direct the reader to the following repository: `https://github.com/tensorflow/models/blob/master/research/slim/nets/nets_factory.py`. We also used the RMC: `deepmind/sonnet/blob/master/sonnet/python/modules/relational_memory.py`

### A.3.4 PPO PARAMETERS

For the projected gym tasks, we used for PPO2 Hyperparameters:

| PPO2 Hyperparameters | Values |
| --- | --- |
| nsteps | 2048 |
| nenvs | 16 |
| nminibatches | 64 |
| $\lambda$ | 0.95 |
| $\gamma$ | 0.99 |
| noptepochs | 10 |
| entropy | 0.0 |
| learning rate | $3 \cdot 10^{-4}$ |
| vf coeffiicent | 0.5 |
| max-grad-norm | 0.5 |
| total time steps | Varying |

See (Cobbe et al., 2018) for the default parameters used for CoinRun. We only varied nminibatches in order to fit memory onto GPU. We also did not use RNN additions, in order to measure performance only from the feedforward network - the framestacking/temporal aspect is replaced by the option to present the agent velocity in the image.

## A.4 THEORETICAL (LQR)

In this section, we use notation consistent with (Fazel et al., 2018) for our base proofs. However, in order to avoid confusion with a high dimensional policy $K$ we described in 3.1, we denote our low dimensional base policy as $P$ and state as $s_t$ rather than $x_t$.

### A.4.1 NOTATION AND SETTING

Let $\|\cdot\|$ be the spectral norm of a matrix (i.e. largest singular value). Suppose $C(P)$ was the infinite horizon cost for an $(A, B, Q, R)$-LQR where action $a_t = -P \cdot s_t$, $s_t$ is the state at time $t$, state transition is $s_{t+1} = A \cdot s_t + B \cdot a_t$, and timestep cost is $s_t^T Q s_t + a_t^T R a_t$.

$C(P)$ for an infinite horizon LQR, while known to be non-convex, still possess the property that when $\nabla C(P^*) = 0$, $P^*$ is a global minimizer, or the problem statement is rank deficient. To ensure that our cost $C(P)$ always remains finite, we restrict our analysis when $P \in \mathcal{P}$, where $\mathcal{P} = \{P : \|P\| \leq \alpha \text{ and } \|A - BP\| \leq 1\}$ for some constant $\alpha$, by choosing $A, B$ and the initialization of $P$ appropriately, using the hyperparameters found in A.3.1. We further define the observation modified cost as $C(K; W_\theta) = C\left(K \begin{bmatrix} W_c \\ W_\theta \end{bmatrix}^\top\right)$.

### A.4.2 SMOOTHNESS BOUNDS

As described in Lemma 16 of (Fazel et al., 2018), we define

$$T_P(X) = \sum_{t=0}^{\infty} (A - BP)^t X [(A - BP)^T]^t \tag{6}$$

and $\|T_P\| = \sup_X \frac{T_P(X)}{\|X\|}$ over all non-zero symmetric matrices $X$.

Lemma 27 of (Fazel et al., 2018) provides a bound on the difference $C(P') - C(P)$ for two different policies $P, P'$ when LQR parameters $A, B, Q, R$ are fixed. During the derivation, it states that when $\|P - P'\| \leq \min\left(\frac{\sigma_{min}(Q)\mu}{4C(P)\|B\|(\|A-BP\|+1)}, \|P\|\right)$, then:

$$C(P') - C(P) \leq 2\|T_P\|\left(2\|P\|\|R\|\|P' - P\| + \|R\|\|P' - P\|^2\right) + \tag{7}$$
$$2\|T_P\|^2 2\|B\|(\|A - BP\| + 1)\|P - P'\|\|P\|^2\|R\|$$

Lemma 17 also states that:

$$\|T_P\| \leq \frac{C(P)}{\mu \sigma_{min}(Q)} \tag{8}$$

where

$$\mu = \sigma_{min}(\mathbb{E}_{x_0 \sim D}[x_0 x_0^T]) \tag{9}$$

Assuming that in our problem setup, $x_0, Q, R, A, B$ were fixed, this means many of the parameters in the bounds are constant, and thus we conclude:

$$C(P') - C(P) \leq \mathcal{O}\left(C(P)^2 \left[\|P\|^2 \|P - P'\| (\|A - BP\| + \|B\| + 1) + \|P\| \|P - P'\|^2\right]\right) \tag{10}$$

Since we assumed $\|A - BP\| \leq 1$ or else $T_P(X)$ is infinite, we thus collect the terms:

$$C(P') - C(P) \leq \mathcal{O}\left(C(P)^2 \left[\|P\|^2 \|P - P'\| + \|P\| \|P - P'\|^2\right]\right) \tag{11}$$

Since $\alpha$ is a bound on $\|P\|$ for $P \in \mathcal{P}$, note that

$$\|P\|^2 \|P - P'\| + \|P\| \|P - P'\|^2 = \|P - P'\| (\|P\|^2 + \|P\| + \|P - P'\|) \tag{12}$$

$$\leq \|P - P'\| (\|P\|^2 + \|P\| (\|P\| + \|P'\|)) \leq (3\alpha^2) \|P - P'\| \tag{13}$$

From (11), this leads to the bound:

$$C(P') - C(P) \leq \mathcal{O}\left(C(P)^2 \|P - P'\|\right) \tag{14}$$

Note that this directly implies a similar bound in the high dimensional observation case - in particular, if $P = K \begin{bmatrix} W_c \\ W_\theta \end{bmatrix}^T$ and $P' = K \begin{bmatrix} W_c \\ W_\theta \end{bmatrix}^T$ then $\|P - P'\| \leq \|K - K'\| \left\|\begin{bmatrix} W_c \\ W_\theta \end{bmatrix}^T\right\| = \|K - K'\|$.

### A.4.3  GRADIENT DYNAMICS IN 1-STEP LQR

We first start with a convex cost 1-step LQR toy example under this regime, which shows that linear components such as $\beta \begin{bmatrix} 0 \\ W_\theta \end{bmatrix}^T$ cannot be removed from the policy by gradient descent dynamics to improve generalization. To shorten notation, let $W_c \in \mathbb{R}^{n \times n}$ and $W_\theta \in \mathbb{R}^{p \times n}$, where $n \ll p$. This is equivalent to setting $d_{signal} = d_{state} = n$ and $d_{noise} = p$, and thus the policy $K \in \mathbb{R}^{n \times (n+p)}$.

In the 1-step LQR, we allow $s_0 \sim \mathcal{N}(0, I)$, $a_0 = K \begin{bmatrix} W_c \\ W_\theta \end{bmatrix} s_0$ and $s_1 = s_0 + a_0$ with cost $\frac{1}{2} \|s_1\|^2$, then

$$C(K; W_\theta) = \mathbb{E}_{s_0}\left[\frac{1}{2} \left\|x_0 + K \begin{bmatrix} W_c \\ W_\theta \end{bmatrix} x_0\right\|^2\right] = \frac{1}{2} \left\|I + K \begin{bmatrix} W_c \\ W_\theta \end{bmatrix}\right\|_F^2 \tag{15}$$

and

$$\nabla C(K; W_\theta) = \left(I + K \begin{bmatrix} W_c \\ W_\theta \end{bmatrix}\right) \begin{bmatrix} W_c \\ W_\theta \end{bmatrix}^T . \tag{16}$$

Define the population cost as $C(K) := \mathbb{E}_{W_\theta}[C(K; W_\theta)]$. Let the notation $O(p, n)$ denote the following set of orthogonal matrices:

$$O(p, n) = \{W \in \mathbb{R}^{p \times n} : W^T W = I\} .$$

We use the shorthand $O(n) = O(n, n)$.

**Proposition 1.** *Suppose that $W_\theta \sim \text{Unif}(O(p, n))$ and $W_c \sim \text{Unif}(O(n))$. Then*

*(i) The minimizer of $C(K)$ is unique and given by $K_\star = \begin{bmatrix} -W_c^T & 0 \end{bmatrix}$.*

*(ii) Thus, the minimizer cost is $C(K_\star) = 0$.*

*Proof.* By standard properties of the Haar measure on $O(p, n)$, we have that $\mathbb{E}[W_\theta] = 0$ and $\mathbb{E}[W_\theta W_\theta^\intercal] = \frac{n}{p} I$. Therefore,

$$
\begin{aligned}
C(K) &= \frac{n}{2} + \frac{1}{2}\mathbb{E}\left[\left\|K\begin{bmatrix} W_c \\ W_\theta \end{bmatrix}\right\|_F^2\right] + \mathbb{E}\left[\operatorname{tr}\left(K\begin{bmatrix} W_c \\ W_\theta \end{bmatrix}\right)\right] \\
&= \frac{n}{2} + \frac{1}{2}\operatorname{tr}\left(K^\intercal K\begin{bmatrix} I & 0 \\ 0 & \frac{n}{p}I \end{bmatrix}\right) + \operatorname{tr}\left(K\begin{bmatrix} W_c \\ 0 \end{bmatrix}\right).
\end{aligned}
$$

We can now differentiate $C(K)$:

$$
\nabla C(K) = K\begin{bmatrix} I & 0 \\ 0 & \frac{n}{p}I \end{bmatrix} + \begin{bmatrix} W_c^\intercal & 0 \end{bmatrix}.
$$

Both claims now follow. $\qquad\square$

### A.4.3.1 Finite Sample Generalization Gap

As an instructive example, we consider the case when we only possess one sample $W_\theta$. Note that if $K = K' + \beta\begin{bmatrix} 0 \\ W_\theta \end{bmatrix}^\intercal$, then $\nabla C(K; W_\theta) = \nabla C(K'; W_\theta) + \beta\begin{bmatrix} 0 \\ W_\theta \end{bmatrix}^\intercal$. In particular, if we perform gradient descent dynamics $K_{t+1} = K_t - \eta\nabla C(K_t; W_\theta)$, then we have

$$
K_t = K_0(I - \eta M)^t + B(I - \eta M)^{t-1} + \dots + B \tag{17}
$$

where $M = \begin{bmatrix} W_c \\ W_\theta \end{bmatrix}\begin{bmatrix} W_c \\ W_\theta \end{bmatrix}^\intercal$ is the Hessian of $C(K; W_\theta)$ and $B = -\eta\begin{bmatrix} W_c \\ W_\theta \end{bmatrix}$. Note that $M$ has rank at most $n \ll n + p$, and thus at a high level, $K_0(I - \eta M)^t$ does not diminish if some portion of $K_0$ lies in the orthogonal complement of the range of $M$. If the initialization is $K_0 \sim \mathcal{N}(0, I)$, then it is highly likely we can find a subspace $Q \subseteq \operatorname{range}(M)^\perp$ for which $K_0 Q \neq 0$.

This is a specific example of the general case where if the Hessian of a function $f(x)$ is degenerate everywhere (e.g. has rank $k < n$), then an $x_0$ initialized with e.g. an isotropic Gaussian distribution cannot converge under gradient descent to a minimizer that lives in the span of the Hessian, as the non-degenerate components do not change. In particular, Proposition 4.7 in (Vershynin, 2009) points to the exact magnitude of the non-degenerate component in the relevant subspace $Q$: $\mathbb{E}_{x\sim\mathcal{N}(0,I)}\left[\left\|\operatorname{Proj}_Q(x)\right\|^2\right] = \frac{n-k}{n}$.

The generalization gap may decrease if the number of level samples is high enough. This can be seen by the sample Hessian of $\widehat{C}(K) = \frac{1}{m}\sum_{i=1}^m C(K; W_{\theta_i})$ being $\widehat{M} = \frac{1}{m}\sum_{i=1}^m M_i$ where $M_i = \begin{bmatrix} W_c \\ W_{\theta_i} \end{bmatrix}\begin{bmatrix} W_c \\ W_{\theta_i} \end{bmatrix}^\intercal \quad \forall i$. In particular, as $m$ increases, the rank of $\widehat{M}$ increases, which allows gradient descent to recover the minimizer $K_\star$ better.

We calculate the exact error of gradient descent on $m$ samples of the 1-step LQR problem.

To simplify notation, we rewrite $W_{\theta_i}$ as $W_i$, and interchangeably use the abbreviations for the finite $C_m(K) = C_m(\cdot; \{W_i\}) = \frac{1}{m}\sum_{i=1}^m C(K; W_i)$ when necessary. We consider gradient descent $K_{t+1} = K_t - \eta\nabla C_m(K_t)$ starting from a random $K_0$ with each entry iid $\mathcal{N}(0, \psi^2)$ and $\eta$ sufficiently small so that gradient descent converges. Let $K_\infty$ denote the limit point of gradient descent. We prove the following generalization gap between $K_\infty$ and $K_{\text{opt}}$.

**Theorem 1.** *Suppose that $n$ divides $p$. Fix an $m \in \{1, ..., p/n\}$. Suppose that the samples $\{W_1, ..., W_m\}$ are generated by first sampling a $W \sim \operatorname{Unif}(O(p))$, and then partitioning the first $n \cdot m$ columns of $W$ into $W_1, ..., W_m$. Suppose gradient descent is run with a sufficiently small step size $\eta$ so that it converges to a unique limit point $K_\infty$. The limit point $K_\infty$ has the following generalization error:*

$$
\mathbb{E}[C(K_\infty)] = \underbrace{\frac{n}{2} + \frac{m^2 n}{2(m+1)^2} + \frac{n^2 m}{2p(m+1)^2} - \frac{m}{m+1}n}_{=:E_1} + \underbrace{\frac{\psi^2 n^2}{2}\left(\frac{m+2}{m+1} - \frac{m^2}{m+1}\frac{n}{p}\right)}_{=:E_2}.
$$

*Here, the expectation is over the randomness of the samples $\{W_i\}_{i=1}^m$ and the initalization $K_0$. The contribution from $E_1$ is due to the generalization error of the* minimum-norm *stationary point of $C_m(\cdot;\{W_i\})$. The contribution from $E_2$ is due to the full-rank initialization of $K_0$.*

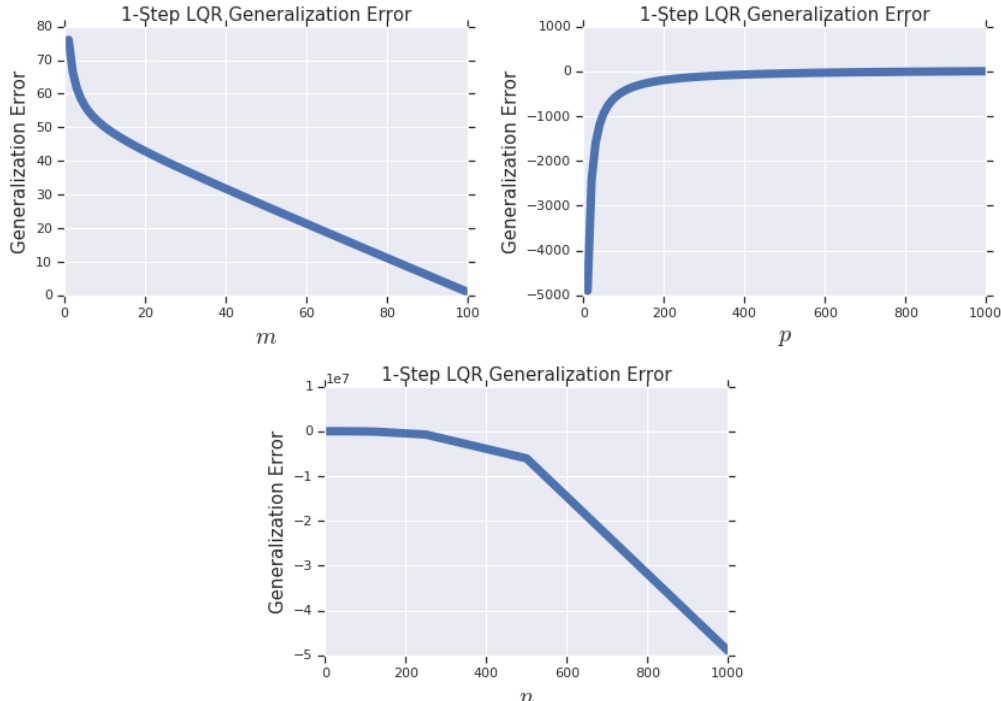

Figure A7: Plot of $\mathbb{E}[C(K_\infty)]$ as a function of $m, p, n$ with elsewhere fixed default values $n = 10$, $p = 1000$, $m = 10$, and $\psi = 1$.

We remark that our proof specifically relies on the rank of the Hessian as $m$ increases, rather than a more common concentration inequality used in empirical risk minimization arguments, which leads to a $\frac{1}{\sqrt{m}}$ scaling. Furthermore, the above expression for $\mathbb{E}[C(K_\infty)]$ does not scale increasingly with $poly(p)$ for the convex 1-Step LQR case, while empirically, the non-convex infinite LQR case does indeed increase from increasing the noise dimension $p$ (as shown in Section 3.1). Interestingly, this suggests that there is an extra contribution from the non-convexity of the cost, where the observation-modified gradient dynamics tends to reach worse optima.

### A.4.3.2    PROOF OF THEOREM 1

Fix integers $n, p$ with $p \geq n$ and suppose that $n$ divides $p$. Draw a random $W \in O(p)$ uniformly from the Haar measure on $O(p)$ and divide $W$ column-wise into $W_1, W_2, ..., W_{p/n}$ (that is $W_i \in \mathbb{R}^{p \times n}$, $W_i^\mathsf{T} W_i = I_n$ and $W_i^\mathsf{T} W_j = 0$ when $i \neq j$). Also draw a $W_c \in O(n)$ uniformly and independent from $W$. Now we consider the matrix $Z_m \in \mathbb{R}^{m \cdot n \times (n+p)}$ for $m = 1, ..., p/n$ defined as:

$$Z_m := \begin{bmatrix} W_c^\mathsf{T} & W_1^\mathsf{T} \\ \vdots & \vdots \\ W_c^\mathsf{T} & W_m^\mathsf{T} \end{bmatrix}.$$

**Proposition 2.** *We have that $Z_m^\dagger$ is given as:*

$$Z_m^\dagger = \begin{bmatrix} \frac{1}{m+1} W_c & \cdots & \frac{1}{m+1} W_c \\ \frac{m}{m+1} W_1 - \frac{1}{m+1} \sum_{i \neq 1} W_i & \cdots & \frac{m}{m+1} W_m - \frac{1}{m+1} \sum_{i \neq m} W_i \end{bmatrix}.$$

*Proof.* Using the fact that $W_c^\mathsf{T} W_c = I$, $W_i^\mathsf{T} W_i = I_n$, and $W_i^\mathsf{T} W_j = 0$ for $i \neq j$, we can compute:

$$Z_m Z_m^\mathsf{T} = I_{m \cdot n} + \begin{bmatrix} I_n \\ \vdots \\ I_n \end{bmatrix} \begin{bmatrix} I_n \\ \vdots \\ I_n \end{bmatrix}^\mathsf{T} .$$

By the matrix inversion formula we can compute the inverse $(Z_m Z_m^\mathsf{T})^{-1}$ as the $m \times m$ block matrix where the diagonal blocks are $\frac{m}{m+1} I$ and the off-diagonal blocks are $-\frac{1}{m+1} I$. We can now write $Z_m^\dagger$ using the formula:

$$Z_m^\dagger = Z_m^\mathsf{T} (Z_m Z_m^\mathsf{T})^{-1} .$$

The result is seen to be:

$$\begin{bmatrix} \frac{1}{m+1} W_c & \cdots & \frac{1}{i+1} W_c \\ \frac{m}{m+1} W_1 - \frac{1}{m+1} \sum_{i \neq 1} W_i & \cdots & \frac{m}{m+1} W_m - \frac{1}{m+1} \sum_{i \neq m} W_i \end{bmatrix} .$$

$\square$

**Proposition 3.** *We have that:*

$$P_{Z_m^\mathsf{T}} = \begin{bmatrix} \frac{m}{m+1} W_c W_c^\mathsf{T} & \frac{1}{m+1} \sum_{i=1}^m W_c W_i^\mathsf{T} \\ \frac{1}{m+1} \sum_{i=1}^m W_i W_c^\mathsf{T} & \frac{m}{m+1} \sum_{i=1}^m W_i W_i^\mathsf{T} - \frac{1}{m+1} \sum_{i \neq j} W_i W_j^\mathsf{T} \end{bmatrix} .$$

*Proof.* We use the formula $P_{Z_m^\mathsf{T}} = Z_m^\mathsf{T} (Z_m Z_m^\mathsf{T})^{-1} Z_m$, combined with the calculations in the previous proposition. $\square$

Now we consider the gradient of $C_m$:

$$\nabla C_m(K) = \frac{1}{m} \sum_{i=1}^m \begin{bmatrix} W_c \\ W_i \end{bmatrix}^\mathsf{T} + K \left( \frac{1}{m} \sum_{i=1}^m \begin{bmatrix} W_c \\ W_i \end{bmatrix} \begin{bmatrix} W_c \\ W_i \end{bmatrix}^\mathsf{T} \right) .$$

Setting $\nabla C_m(K) = 0$, we see that minimizers of $C_m$ are solutions to:

$$K \left( \sum_{i=1}^m \begin{bmatrix} W_c \\ W_i \end{bmatrix} \begin{bmatrix} W_c \\ W_i \end{bmatrix}^\mathsf{T} \right) = -\sum_{i=1}^m \begin{bmatrix} W_c \\ W_i \end{bmatrix}^\mathsf{T} .$$

Using our notation above, this is the same as:

$$K Z_m^\mathsf{T} Z_m = -\begin{bmatrix} I_n & \cdots & I_n \end{bmatrix} Z_m .$$

The minimum norm solution, denoted $K_{\min}$, to this equation is:

$$K_{\min} = -\begin{bmatrix} I_n & \cdots & I_n \end{bmatrix} (Z_m^\dagger)^\mathsf{T} .$$

Next, we recall that $C(K) = \mathbb{E}[C_m(K)]$ is:

$$C(K) = \frac{n}{2} + \frac{1}{2} \operatorname{tr} \left( K^\mathsf{T} K \begin{bmatrix} I & 0 \\ 0 & \frac{n}{p} I \end{bmatrix} \right) + \operatorname{tr} \left( K \begin{bmatrix} W_c \\ 0 \end{bmatrix} \right) .$$

Our next goal is to compute $\mathbb{E}[C(K_{\min})]$. First, we observe that:

$$\begin{aligned} \operatorname{tr} \left( K_{\min} \begin{bmatrix} W_c \\ 0 \end{bmatrix} \right) &= -\left\langle Z_m^\dagger, \begin{bmatrix} W_c & \cdots & W_c \\ 0 & \cdots & 0 \end{bmatrix} \right\rangle \\ &= -\frac{m}{m+1} \operatorname{tr}(W_c^\mathsf{T} W_c) \\ &= -\frac{m}{m+1} n . \end{aligned}$$

Next, defining $B_i := \frac{m}{m+1} W_i - \frac{1}{m+1} \sum_{j \neq i} W_j$, we compute $K_{\min}^\mathsf{T} K_{\min}$ as:

$$
K_{\min}^\mathsf{T} K_{\min} = \begin{bmatrix} \frac{1}{m+1} W_c & \cdots & \frac{1}{m+1} W_c \\ B_1 & \cdots & B_m \end{bmatrix} \begin{bmatrix} I & \cdots & I \\ \vdots & \ddots & \vdots \\ I & \cdots & I \end{bmatrix} \begin{bmatrix} \frac{1}{m+1} W_c^\mathsf{T} & B_1^\mathsf{T} \\ \vdots & \vdots \\ \frac{1}{m+1} W_c^\mathsf{T} & B_m^\mathsf{T} \end{bmatrix}
$$

$$
= \begin{bmatrix} \frac{m^2}{(m+1)^2} I & * \\ * & (\sum_{i=1}^m B_i)(\sum_{i=1}^m B_i)^\mathsf{T} \end{bmatrix}
$$

$$
= \begin{bmatrix} \frac{m^2}{(m+1)^2} I & * \\ * & \frac{1}{(m+1)^2} (\sum_{i=1}^m W_i)(\sum_{i=1}^m W_i)^\mathsf{T} \end{bmatrix} .
$$

Therefore:

$$
\mathrm{tr}\left( K_{\min}^\mathsf{T} K_{\min} \begin{bmatrix} I & 0 \\ 0 & \frac{n}{p} I \end{bmatrix} \right) = \frac{m^2}{(m+1)^2} n + \frac{n}{p(m+1)^2} (nm) .
$$

Putting everything together, we have that:

$$
C(K_{\min}) = \frac{n}{2} + \frac{m^2 n}{2(m+1)^2} + \frac{n^2 m}{2p(m+1)^2} - \frac{m}{m+1} n .
$$

Notice that this final value is not a function of the actual realization of $W$.

We now consider the second source of error, which comes from the initialization $K_0$. Recall that each entry of $K_0$ is drawn iid from $\mathcal{N}(0, \psi^2)$.

Let us consider the gradient descent dynamics:

$$
K_{t+1} = K_t - \eta \nabla C_m(K_t) .
$$

Unrolling the dynamics under our notation:

$$
K_t = K_0 (I - (\eta/m) Z_m^\mathsf{T} Z_m)^t - (\eta/m) \begin{bmatrix} I_n & \cdots & I_n \end{bmatrix} Z_m \sum_{k=0}^{t-1} (I - (\eta/m) Z_m^\mathsf{T} Z_m)^k .
$$

It is not hard to see that for $\eta$ sufficiently small, as $t \to \infty$

$$
\lim_{t \to \infty} -(\eta/m) \begin{bmatrix} I_n & \cdots & I_n \end{bmatrix} Z_m \sum_{k=0}^{t-1} (I - (\eta/m) Z_m^\mathsf{T} Z_m)^k = K_{\min} .
$$

Hence:

$$
K_\infty = K_0 (I - (\eta/m) Z_m^\mathsf{T} Z_m)^\infty + K_{\min} .
$$

Call the matrix $M_\infty := (I - (\eta/m) Z_m^\mathsf{T} Z_m)^\infty$. We observe that:

$$
C(K_\infty) = \frac{n}{2} + \frac{1}{2} \mathrm{tr}\left( M_\infty^\mathsf{T} K_0^\mathsf{T} K_0 M_\infty \begin{bmatrix} I & 0 \\ 0 & \frac{n}{p} I \end{bmatrix} \right)
$$

$$
+ \frac{1}{2} \mathrm{tr}\left( (M_\infty^\mathsf{T} K_0^\mathsf{T} K_{\min} + K_{\min}^\mathsf{T} K_0 M_\infty) \begin{bmatrix} I & 0 \\ 0 & \frac{n}{p} I \end{bmatrix} \right)
$$

$$
+ \frac{1}{2} \mathrm{tr}\left( K_{\min}^\mathsf{T} K_{\min} \begin{bmatrix} I & 0 \\ 0 & \frac{n}{p} I \end{bmatrix} \right)
$$

$$
+ \mathrm{tr}\left( K_0 M_\infty \begin{bmatrix} W_c \\ 0 \end{bmatrix} \right)
$$

$$
+ \mathrm{tr}\left( K_{\min} M_\infty \begin{bmatrix} W_c \\ 0 \end{bmatrix} \right) .
$$

Noticing that $K_0$ is independent of $Z_m$, taking expectations

$$
\mathbb{E}[C(K_\infty)] = \mathbb{E}[C(K_{\min})] + \frac{1}{2} \mathbb{E} \, \mathrm{tr}\left( M_\infty^\mathsf{T} K_0^\mathsf{T} K_0 M_\infty \begin{bmatrix} I & 0 \\ 0 & \frac{n}{p} I \end{bmatrix} \right)
$$

$$
= \mathbb{E}[C(K_{\min})] + \frac{\psi^2 n}{2} \mathbb{E} \, \mathrm{tr}\left( M_\infty^\mathsf{T} M_\infty \begin{bmatrix} I & 0 \\ 0 & \frac{n}{p} I \end{bmatrix} \right) .
$$

Above, we use the fact that $\mathbb{E}[K_0^\mathsf{T} K_0] = \psi^2 n I_{n+p}$. Now it is not hard to see that for $\eta$ sufficiently small, we have that $M_\infty = I - P_{Z_m^\mathsf{T}}$. On the other hand, using $\mathbb{E}[W_i W_j^\mathsf{T}] = 0$, we have that

$$
\mathbb{E}[P_{Z_m^\mathsf{T}}] = \begin{bmatrix} \frac{m}{m+1}I & 0 \\ 0 & \frac{m^2}{m+1}\frac{n}{p}I \end{bmatrix} .
$$

Therefore:

$$
\mathbb{E}[M_\infty^\mathsf{T} M_\infty] = \begin{bmatrix} (1 - m/(m+1))I & 0 \\ 0 & (1 - m^2 n/((m+1)p)I \end{bmatrix} .
$$

Plugging in to the previous calculations, this yields

$$
\mathbb{E}[C(K_\infty)] = \mathbb{E}[C(K_{\min})] + \frac{\psi^2 n^2}{2}\left( \frac{m+2}{m+1} - \frac{m^2}{m+1}\frac{n}{p} \right) .
$$

