# OpenReview forum: "Observational Overfitting in Reinforcement Learning"
_ICLR.cc/2020/Conference — Accept (Poster)_

### Official Review · AnonReviewer2 · 2019-10-15
**Official Blind Review #2**

**Rating:** 8

**Review:**

Claims: This paper studies instances where observational overfitting is occurring in reinforcement learning, and separates it from other confounding factors contributing to poor generalization. They specifically study observational overfitting with LQR and deep neural networks, and show implicit regularization naturally occurs in this setting.

Decision: Accept. The paper presents relevant bounds on overfitting depending on the dimensionality of the signal and noise. The framework and definition of family of MDPs are interesting and useful, and the focus on the implicit regularization provided by overparameterization when dealing with high dimensional observations with correlated noise is important and useful to decouple from varying dynamics.

Section 3.2 needs more explanation of the experiment, how the train test split is created and if this is on the original rendered images from OpenAI gym or with correlated noise -- I've never seen such stark generalization gaps one the original environments. Are all of these experiments done with the proprioceptive state? Are the experiments in 3.3 then done with learned deconvolutions to generate the 84x84 images? Why this instead of using the rendered images themselves (as f) with added noise parameterized by \theta?  Fig 8 is also not well explained, there are two sets of lines of each color in the plots with no label as to which is which, I assume these are train and test? Perhaps a plot that shows the generalization gap as the y-axis would be more clear.

**Experience Assessment:**

I have read many papers in this area.

**Review Assessment: Checking Correctness Of Derivations And Theory:**

I did not assess the derivations or theory.

**Review Assessment: Checking Correctness Of Experiments:**

I assessed the sensibility of the experiments.

**Review Assessment: Thoroughness In Paper Reading:**

I read the paper at least twice and used my best judgement in assessing the paper.

---

> ### Author Response · Authors · 2019-11-09
> **Author Response to Official Blind Review #2**
>
> Thank you for your review! We’ve added clarifications about our setup in the paper, and to address your specific questions:
>
> ----------Q1. Train/test split: Yes, we do use the same exact observation function sampling as the LQR case to produce a distribution of MDP’s, for which we can sample from to produce train/test sets. So now each sampled Mujoco environment (using seed \theta) has an extra “background” observation vector from the output of g_{\theta}.
>
> A reason that we still project the original images using f is to fairly ensure that all environments have the same output dimensions (30 from f, 100 from g) since the Mujoco environments have different vanilla state dimensions (otherwise different observation dimensions can affect results, as seen from Theorem 3.1 and Figure 3 (left)). Regardless of the vanilla state dimension, to prevent information loss and still allow the new projected MDP to be solvable, we use orthogonal projection matrices to construct f and g.
>
> For 3.3, we use the 1D state vector corresponding to e.g. joint-locations, not the RGB rendered image from env.render(). We wanted to avoid leaps in logic from assuming the underlying state is low dimensional, and instead enforce that the actual state is indeed low-dimensional. Furthermore, requiring inferring necessary state information (such as velocity) from the env.render() image would introduce additional complications (e.g. requiring framestacking, etc.).
>
> We use a very similar (f,g) - projection method on this 1D state vector, pictorially explained in Figure 2(b) where the 2 image outputs from (f,g) are then concatenated by halving one of each. In terms of how we project a 1D state to an “image”, we perform the same method for both f and g:
> We first project the (1D) vanilla Gym state to a larger 1D latent using an orthogonal matrix.
> We reshape this latent to a 2D square (which also has an extra filter dimension), and send this square through deconvolutions with orthogonally sampled weight matrices (specified in Appendix 3.2) for (f,g). In particular, the final image size is the original 84x84x3 used for Atari and NatureCNN.
>
> ----------Q2. CoinRun curves (Fig 8): We wrote in the caption to clarify that the curves are indeed training vs testing. One aspect of the benchmark CoinRun is that the training set contains “easier” levels to help the agent burn-in good initial behavior due to its sparse reward regime, while the test set is entirely “difficult” levels - unfortunately showing only the difference between train and test would not make sense here as they are not strictly from the same distribution. However, we believe that it is intriguing that even under these conditions, the improved generalization effect of overparametrization still holds.

---

### Official Review · AnonReviewer3 · 2019-10-16
**Official Blind Review #3**

**Rating:** 8

**Review:**

Summary:

The paper proposes a method for measuring generalization error in reinforcement learning (RL). The paper proposes to disentangle the observations into the features that relevant and not-relevant for the policy and then perturb the non-relevant features while keeping the relevant features constant. If the agent has learned to generalize, then perturbing the non-relevant features should not change the RL score.

Comments:

1. The paper addresses an important question in RL:  generalization in the observational space. It is not trivial to define generalization in RL as this differs fundamentally from a SL or unsupervised learning setting.  The paper proposes a metric to measure this generalization error and this can be applied to non-toyish environments.

2. The paper is clearly written and well-motivated.

3.  I do like the proposed method. However, I also see some shortcomings of the method.
The paper proposes to disentangle the features into relevant and non-relevant features. While this might be easier for certain task, it might be much more difficult for other tasks. The relevant features may be some implicit priors that are not easy to extract, for example, the fundamental physics of an environment. I am not sure how this can be addressed in a complicated environment where the relevant features are sampled from an implicit prior.

4. The experiments on CoinRun i think is the most relevant ones. However, in this environments, it seems that although the observational features are quite different (rendering of the environment), the underlying physics or moves/ actions are very much similar. It would be nice to see a more complicated environment where the underlying physics or composition of actions can be different.

**Experience Assessment:**

I have read many papers in this area.

**Review Assessment: Checking Correctness Of Derivations And Theory:**

I assessed the sensibility of the derivations and theory.

**Review Assessment: Checking Correctness Of Experiments:**

I assessed the sensibility of the experiments.

**Review Assessment: Thoroughness In Paper Reading:**

I read the paper at least twice and used my best judgement in assessing the paper.

---

> ### Author Response · Authors · 2019-11-09
> **Author Response to Official Blind Review #3**
>
> Thank you for your encouraging comments!
>
> ----------Q1: Difficult to disentangle for other tasks
> We do agree that in many benchmarks, one cannot easily split the features - we did so in this work to simplify the setting and make results less ambiguous, more rigorous, and more principled. However, we believe that multiple extensions of this formulation may exist, which could be a promising direction for future work.
>
> For instance, one may perform another linear projection using a new matrix (e.g. W_{final}) after our (f,g)-observation to prevent the inductive bias of “only looking at f”. In a more general sense, the outputs of (f,g) can be thought of as factored latent vectors themselves, which can be sent to a more complicated decoder function (h) to produce an image. In a sense this is the inverse way of thinking about a beta-VAE [1] or InfoGAN [2] whose latents are highly disentangled - this type of reasoning we find is highly relevant to recent works on causal inference [3].
>
> ----------Q2: CoinRun
> Indeed, while the dynamics of CoinRun are similar, they are still strictly different per level (e.g. the height of a wall or position of a monster is randomly sampled per level), and hence we find it interesting that overparameterization results still carry over when dynamics are somewhat different.
>
> For environments where the dynamics are much more different between levels, this would be interesting future work but currently out of the scope of our paper which emphasizes observational overfitting and feed-forward networks. We believe that varying dynamics would have a stronger emphasis on the generalization quality of RNN’s. For instance, [4] analyzes overfitting with procedurally generated mazes (“RandomMazes”) where the observation is much more simple (e.g. each grid/pixel is binary, representing a wall or a floor) than the varying state transitions in the maze. However, the agent still overfits, and the authors focus on only analyzing the LSTM in the policy.
>
> It is certainly an important question how to decide which part of the policy is overfitting more and thus when to regularize the RNN or CNN more, when dealing with both varying observations and dynamics. It is also an open question whether certain implicit regularizations occur for RNN’s especially in this RL regime. Some results for SL exist: [5] asserts that Path-SGD improves RNN generalization performance on Sequential MNIST.
>
>
>
> [1] Irina Higgins et al. beta-VAE: Learning Basic Visual Concepts with a Constrained Variational Framework, ICLR 2017.
> [2] Xi Chen, Yan Duan, Rein Houthooft, John Schulman, Ilya Sutskever, Pieter Abbeel. InfoGAN: Interpretable Representation Learning by Information Maximizing Generative Adversarial Nets, NeurIPS 2016.
> [3] Martin Arjovsky, Léon Bottou, Ishaan Gulrajani, David Lopez-Paz. Invariant Risk Minimization, arXiv:1907.02893, 2019.
> [4] Karl Cobbe et al. Quantifying Generalization in Reinforcement Learning, ICML 2019.
> [5] Behnam Neyshabur. Implicit Regularization in Deep Learning. PhD Thesis, 2017.

---

### Official Review · AnonReviewer1 · 2019-10-23
**Official Blind Review #1**

**Rating:** 6

**Review:**

Review:

This paper considers the problem of overfitting in RL through a specific model of overfitting, namely one where noise is introduced in the observation space, independently of the controllable dynamics.  The paper provides both theoretical and empirical insights into the various manifestations of overfitting in this class of domains.

A strength of this paper is to deepen our understanding of the phenomenon of overfitting in RL.  To get a deep understanding of hard problems it is worthwhile to look at sub-classes of problems, and as such the identification of the observational overfitting setting is interesting.  The paper provides insightful findings such as:
-	Explicit separation of f and g_theta.
-	Theoretical properties of generalization for the specific case of LQR and linear policies.
There are also several empirical results, on three contrasting domains which illustrates various aspects of the interaction between parameterization vs generalization.

But overall, though I read the paper in detail, and am knowledgeable of the topic, I am still struggling with extracting very specific new findings from this work.  Here are a few examples:
p.6:  “We observe empirically that the underlying state dynamics has a significant effect on generalization performance as the policy nontrivially increased test performance” ->  I see where you draw this analysis from the results.  But is this finding new or surprising?  I would have been very comfortable saying (without your results) that the underlying state dynamics can have a significant effect on generalization performance.
p.6: “This suggests that the Rademacher complexity and the weight perturbation bound for rewards vary highly for different environments.”  -> Again, how is this new knowledge?
p.7: “this also suggests that the RL generalization quality of a convolutional architecture is not limited to real world data” -> Same question, what new knowledge have we gained?

Perhaps the key message is that “there are generalization benefits to overparameterization” and that “implicit generalization” plays a key role in controlling this?   If that is the main message, than it is somewhat obfuscated by all the material on LQR, which doesn’t really go in this direction (p.4: “We being with a theorem which implies that a high dimensional observational space directly contributes to overfitting”).  I must say I found most of the material in 3.1 to distrct from the main message – but perhaps it is because I am not very interested in the LQR setting, and did not understand how the findings there supported those in latter sections.

Overall, I would say the paper suffers from some clarity issues.  There are some minor typos, then some key terms of are not sufficiently explained/defined (e.g. g_\theta in Sec.2.2 is said to project “the latent data to unimportant features”, but then there is discussion of “in settings where $g$ does matter” – I’m confused.)   It’s not clear whether Thm 3.1 is new, or adaptation of existing results.  More broadly, I found Sec.3.1 difficult to follow. The last paragraph of Sec.4 seems superfluous.   I would recommend some good editing throughout.

Another concern with the work is the fact that several aspects of RL are ignored (e.g. exploration, entropy, \gamma, noise, stochastic gradients, etc. – as per p.4), yet are known to have an effect on overfitting, perhaps much more than the depth and width of the neural network.  If that’s the case, it is perhaps dangerous to ignore them in the analysis; it is not described in the paper, for the domains studied in the empirical results, whether this is the case or not.  As a result, it’s not clear how far the current analysis carries.  I’m also not clear on the analysis at bottom p.7 “IMPALA-LARGE (…) memorizes less than IMPA due its inherent inductive bias” -> Seems to me IMPALA-LARGE might simply be in underfitting regime.

======
Update post-rebuttal:
Thank you for the thoughtful responses.   Given the other strong reviews, and your careful characterization of various aspects of the work, I am favorable to accepting.  Please try to incorporate elements of this discussion into the paper, which I think will give a richer and more nuanced understanding of the work to future readers.

**Experience Assessment:**

I have published in this field for several years.

**Review Assessment: Checking Correctness Of Derivations And Theory:**

I assessed the sensibility of the derivations and theory.

**Review Assessment: Checking Correctness Of Experiments:**

I assessed the sensibility of the experiments.

**Review Assessment: Thoroughness In Paper Reading:**

I read the paper at least twice and used my best judgement in assessing the paper.

---

> ### Author Response · Authors · 2019-11-09
> **Author Response to Official Blind Review #1, Part 1**
>
> Thank you for your feedback. We revised the paper to make the conclusion, contributions, and experiment setup clearer. In terms of the observational overfitting part, we agree that some of our settings may seem conceptually simple - e.g. if put into words, phrases such as “an agent overfits to the background” or “state dynamics affects generalization gap” sound obvious. However, we believe that there has not been a rigorous study on how exactly these effects happen, or what types of tools (SL generalization bounds, Rademacher complexity, margin distributions, etc.) we can use to empirically analyze these effects. One of the main benefits of this work is to explicitly analyze an effect common to most RL generalization works involving non-trivial observation spaces, and we hope that this provides much needed clarity and guidance for evaluating RL generalization. We agree that there have been numerous previous works about the "background effect" in RL, which focus on augmenting 2D images with new colors or shapes in the background and thus this is not a new concept. However, we have shown that observational overfitting occurs even for 1D states from using linear projections, and we thus provide a common and rigorous unifying theme here.
>
> We also believe that the effect of implicit regularization in RL is non-obvious. Our empirical results, such as high width/layers on Mujoco improving generalization, are particularly surprising and non-obvious as normally only basic 2-layer MLPs are used as policies. Also, the idea of using overparametrization in order to improve generalization is novel in RL, and suggests that implicit regularization is a common factor among both RL and SL theory that should be studied more.
>
> Below are our extended responses to your concerns. If after looking at the revision and our response you believe that we have addressed your concerns well and our paper is interesting enough to be accepted at ICLR, we hope that you would kindly increase your rating.
>
> ####################### DETAILED RESPONSES ###########################
>
> ---------- Q1: Extracting new findings
> To address your specific questions:
> 1. Underlying state dynamics have effects on generalization and Rademacher complexity - What we meant was that while one can produce generalization bounds from smoothness bounds on Reward(\pi) - Reward(\pi’), this is difficult to do for Mujoco agents as the physics simulator’s state dynamics is highly complex. We instead can observe the differences in generalization gaps by looking at varying agents (Swimmer, Halfcheetah, etc.) with different state transition dynamics.
>
> 2. Generalization of convolution not limited to real world data - A large number of works design their environments with significant emphasis on “real world” backgrounds [1,2,3]. Using our synthetic construction which simply uses linear projections from the latent state, we find that this may not be necessary. We showed that generalization ranking among NatureCNN/IMPALA/IMPALA-LARGE remains the same regardless of whether we use our synthetic construction or CoinRun, which suggests that the generalization performance is perhaps due to some common factors. This leads to our memorization experiments, which supports the claim that implicit regularization is one of these factors, as it shows a ConvNet with more parameters to memorize less. This is supported by the the memorization performance decrease in Figure 7 and the more discretized memorization limits from Figure A4 in the Appendix.

---

> > ### Author Response · Authors · 2019-11-09
> > **Author Response to Official Blind Review #1, Part 2**
> >
> > ---------- Q2: Material on LQR
> > We agree that LQR has its limitations for studying deep RL. However, we also believe it is a critical first step to analyze the base case of deep RL. One reason is because LQR has provided numerous insights into control theory ranging from sample complexity [4] to model-based RL [5].
> >
> > Furthermore, in general, it is typical to study a simpler theoretical problem to visualize what may occur in deep learning [8,9,10], as it can both introduce profound theoretical insight and avoids certain hyperparameter effects that may confound the results (which partially addresses your hyperparameter question). Study on linear/logistic regression [6] has resulted in important tools to study deep supervised learning [7].
> >
> > There are multiple benefits to analyzing the LQR case, specifically:
> > ------1. The LQR case is especially important as it allows deterministic exact gradient computation (e.g. the policy can differentiate through the total reward function), whereas common RL requires stochastic gradients and the PPO objective function contains multiple other objectives such as entropy regularization and value function loss. Thus LQR can much more cleanly provide evidence of implicit regularization. Furthermore, the LQR cost is readily a nonconvex function as opposed to the convex loss for linear regression, which may introduce non-trivial effects.
> >
> > ------2. To show that overparametrization alone is an important implicit regularizer in RL, LQR allows us to use linear policies and consequently also stack linear layers, to show that overparametrization alone can affect gradient dynamics, which is not a consequence of extra representation power from extra ReLU/Tanh layers. There have been multiple recent works on this linear-layer stacking in SL and other theoretical problems such as matrix factorization [11,12,13], but to our knowledge we are the first to show this for RL generalization.
> >
> > ------3. Combining points (1,2), LQR is much more practical in terms of checking the predictiveness of various generalization bounds found in SL, especially because it uses deterministic raw outputs. Stochastic actions would significantly complicate this setting because the norm of the policy weights would not directly correspond to the magnitude of the output. For instance, discrete actions (using Gumbel-Softmax) would require margin bounding because the weight norms are not bounded. We perform margin bound analysis on CoinRun in Appendix A2. For continuous actions (e.g. using N(\mu, \sigma)), it is unclear how the norms of the network weights affect \sigma and how this interacts with the total reward. However, we still perform analysis for Gym environments in Section 3.2 + 3.3.
> >
> > ------4. If certain bounds empirically do not predict generalization gaps for LQR using linear layers, then it is unlikely these bounds have hope of predicting the case for when the policy uses nonlinearities such as ReLU or Tanh in stochastic policy gradient. We believe that this is an important result in itself for both RL and SL as it shows that such bounds currently cannot extend to the RL case.
> >
> > To clarify about another key message in our paper: we believe that overparametrization and observational overfitting are coupled. [14] did not find such a result where the larger ConvNet memorized less, while we do when the overfitting is caused by our construction of the observation space, and thus me need to make the distinction that overparametrization does not necessarily always induce implicit regularization in the broader RL regime where dynamics overfitting can occur.
> >
> > ---------- Q3: Exploration, entropy, etc. ignored
> > We agree that those hyperparameters in the RL algorithm can strongly affect generalization. For instance, the gamma can implicitly decrease the hypothesis class of policies [15] and entropy can smoothen the loss landscape [16], and their effects can be quite strong. However, these hyperparameters are not the main message of this work, which is only focused on the observation space. This is particularly why we needed to fix these hyperparameters in the deep learning experiments for an apples-to-apples comparison involving changing architectures, and avoid these hyperparameters altogether in LQR.
> >
> > We also acknowledge that some hyperparameters potentially can complicate the observational overfitting regime - for instance low batch size/replay buffer size can perhaps affect how resistant the network is to the output of the g-function, and a policy with higher entropy may produce more diverse features which affects generalization [17]. To keep the experiments tractable, we picked reasonable (fixed) hyperparameters to minimize these possible effects. For instance, in the Gym + CoinRun benchmarks, our effective batch size was relatively high by using multiple (16+) MPI-workers, and we set entropy = 0.0 for PPO for Mujoco, which is also practically an optimal parameter [18].
> >
> > Thank you for your time!

---

> > > ### Author Response · Authors · 2019-11-09
> > > **Author Response to Official Blind Review #1, Part 3**
> > >
> > > ############################## References ################################
> > >
> > >
> > > [1] Shani Gamrian, Yoav Goldberg. Transfer Learning for Related Reinforcement Learning Tasks via Image-to-Image Translation, ICML 2019.
> > > [2] Amy Zhang, Yuxin Wu, and Joelle Pineau. Natural Environment Benchmarks for Reinforcement Learning, AAAI 2018.
> > > [3] Karl Cobbe et al. Quantifying Generalization in Reinforcement Learning, ICML 2019.
> > > [4] Sarah Dean, Horia Mania, Nikolai Matni, Benjamin Recht, Stephen Tu. On the Sample Complexity of the Linear Quadratic Regulator, arXiv: 1710.01688, 2018.
> > > [5] Stephen Tu, Benjamin Recht. The Gap Between Model-Based and Model-Free Methods on the Linear Quadratic Regulator: An Asymptotic Viewpoint. COLT 2019.
> > > [6] Sham M. Kakade, Karthik Sridharan, Ambuj Tewari. On the Complexity of Linear Prediction:
> > > Risk Bounds, Margin Bounds, and Regularization.  NeurIPS 2009.
> > > [7] Peter Bartlett, Dylan J. Foster, Matus Telgarsky. Spectrally-normalized margin bounds for neural networks, NeurIPS 2017
> > > [8] Constantinos Daskalakis, Andrew Ilyas, Vasilis Syrgkanis, Haoyang Zeng. Training GANs with Optimism, ICLR 2018.
> > > [9] Jerry Li, Aleksander Madry, John Peebles, Ludwig Schmidt, On the Limitations of First-Order Approximation in GAN Dynamics, ICML 2018.
> > > [10] Dimitris Tsipras, Shibani Santurkar, Logan Engstrom, Alexander Turner, Aleksander Madry. Robustness May Be at Odds with Accuracy, ICLR 2019.
> > > [11] Sanjeev Arora, Nadav Cohen, Wei Hu, Yuping Lu. Implicit Regularization in Deep Matrix Factorization, NeurIPS 2019.
> > > [12] Sanjeev Arora, Nadav Cohen, Elad Hazan. On the Optimization of Deep Networks: Implicit Acceleration by Overparameterization, ICML 2018.
> > > [13] Gunasekar, Suriya and Woodworth, Blake E and Bhojanapalli, Srinadh and Neyshabur, Behnam and Srebro, Nati. Implicit regularization in matrix factorization, NeurIPS 2017.
> > > [14] Chiyuan Zhang, Oriol Vinyals, Remi Munos, Samy Bengio. A Study on Overfitting in Deep Reinforcement Learning, arXiv:1804.06893, 2018.
> > > [15] Jiang, Nan and Kulesza, Alex and Singh, Satinder and Lewis, Richard. The Dependence of Effective Planning Horizon on Model Accuracy, AAMAS 2015.
> > > [16] Zafarali Ahmed, Nicolas Le Roux, Mohammad Norouzi, Dale Schuurmans. Understanding the impact of entropy on policy optimization, ICML 2019.
> > > [17] Maximilian Igl et al. Generalization in Reinforcement Learning with Selective Noise Injection and Information Bottleneck, NeurIPS 2019.
> > > [18] John Schulman et al. Proximal Policy Optimization Algorithms, 2017. Code: https://github.com/openai/baselines/blob/master/baselines/ppo2/ppo2.py#L21

---

### Author Response · Authors · 2019-11-09
**Meta response for all reviewers**

Thank you all for your time in reviewing our paper - we appreciate the feedback and believe all questions have helped us improve the paper.

Changes to paper:
Section 1: Added paper outline
Section 2: Added rigorous example of linear regression in high dimensional space as example of implicit regularization, and highlighted a unifying theme of observational overfitting by discussing the novel 1D state case.
2.1 More rigorous and clear description of observation function and how we generate datasets
2.2 Clarified Rademacher Complexity and purpose of g-function in meta-learning
3.1 Clarified purpose of LQR and construction of setup
3.2 Clarified construction of Gym environment setup
3.3 Clarified logic on convolution generalization performance -> memorization test

---

### Decision · Program_Chairs · 2019-12-19

**Decision:**

Accept (Poster)

**Comment:**

The paper proposes a way to analyze overfitting to non-relevant parts of the state space in RL and proposes a framework to measure this type of generalization error. All reviewers agree that the formulation is interesting and useful for practical RL.